# What Matters in Graph Class Incremental Learning? An Information Preservation Perspective

**Jialu Li**[1,2,3,*]
jialuli@tju.edu.cn

**Yu Wang**[1,2,3,*]
wang.yu@tju.edu.cn

**Pengfei Zhu**[1,2,3,†]
zhupengfei@tju.edu.cn

**Wanyu Lin**[4]
wan-yu.lin@polyu.edu.hk

**Qinghua Hu**[1,2,3]
huqinghua@tju.edu.cn

[1]College of Intelligence and Computing, Tianjin University, Tianjin, China
[2]Engineering Research Center of City Intelligence and Digital Governance,
Ministry of Education of the People's Republic of China, Tianjin, China
[3]Haihe Lab of ITAI, Tianjin, China
[4]Department of Computing, The Hong Kong Polytechnic University, Hong Kong, China

## Abstract

Graph class incremental learning (GCIL) requires the model to classify emerging nodes of new classes while remembering old classes. Existing methods are designed to preserve effective information of old models or graph data to alleviate forgetting, but there is no clear theoretical understanding of what matters in information preservation. In this paper, we consider that present practice suffers from high semantic and structural shifts assessed by two devised shift metrics. We provide insights into information preservation in GCIL and find that maintaining graph information can preserve information of old models in theory to calibrate node semantic and graph structure shifts. We correspond graph information into low-frequency local-global information and high-frequency information in spatial domain. Based on the analysis, we propose a framework, Graph Spatial Information Preservation (GSIP). Specifically, for low-frequency information preservation, the old node representations obtained by inputting replayed nodes into the old model are aligned with the outputs of the node and its neighbors in the new model, and then old and new outputs are globally matched after pooling. For high-frequency information preservation, the new node representations are encouraged to imitate the near-neighbor pair similarity of old node representations. GSIP achieves a 10% increase in terms of the forgetting metric compared to prior methods on large-scale datasets. Our framework can also seamlessly integrate existing replay designs. The code is available through `https://github.com/Jillian555/GSIP`.

## 1 Introduction

In real-world applications, graph data is continuously generated. For instance, in citation networks, new types of papers and their citations may constantly emerge, an ideal literature classifier needs to continuously distinguish literature in emerging research areas [1, 2]. Therefore, it is critical for a graph model to incrementally integrate new classes on an extended graph, which is referred to as Graph class incremental learning (GCIL). However, this poses a major challenge known as catastrophic forgetting, where the model needs to preserve previous information while continuously acquiring new information [3, 4].

---

*Equal contribution.
†Corresponding author.

38th Conference on Neural Information Processing Systems (NeurIPS 2024).

Many approaches attempt to preserve information from previous models or graph data to prevent catastrophic forgetting in GCIL, which can be divided into four groups. The parameter isolation methods entirely or partially preserve parameters of different tasks to protect model performance, such as dynamically incrementing feature extractors and prototypes [2]. Regularization methods, on the one hand, preserve important parameters, such as assessing parameter importance by considering loss and topology [1], and maintaining orthogonality with parameters from previous tasks [5], on the other hand, preserve the absolute position of nodes in feature space or output space, such as aligning the outputs of samples on old and new models [6]. The replay methods preserve a few nodes or subgraphs to retrain the model to prevent forgetting, such as saving representative nodes [7], selecting subgraphs according to node degree [8], and compressing training graphs [9]. The hybrid methods combine different learning paradigms (*i.e.* the combination of replay and regularization methods), such as feature distillation after identifying critical nodes [10] and minimizing distribution disparity of selected nodes across new and prior models [11]. These hybrid methods have demonstrated considerable potential and achieved state-of-the-art results. Despite their effectiveness, the information preservation mechanism by existing methods remains unclear, making it challenging to develop effective solutions for GCIL. This motivates us to explore a fresh perspective: ***What matters in information preservation when learning from the old model to the new model for GCIL?***

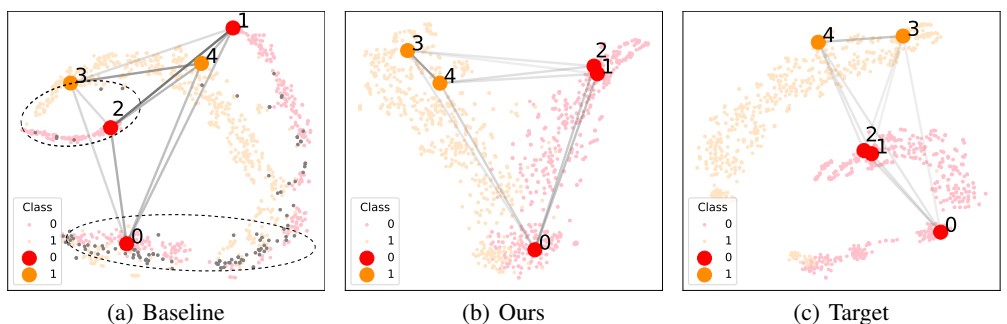

(a) Baseline          (b) Ours          (c) Target

Figure 1: Visualizations of semantic shift and structure shift.

We investigate the unique characteristics of catastrophic forgetting on graphs and find **node semantic and graph structure shifts** in GCIL. The visualization of node embeddings for new model of baseline (ERGNN [7]) and our method on old classes of CoraFull dataset are exhibited in Figure 1. The structure learned by old model is selected as target instead of original topology due to noisy real structure [12]. Five nodes belonging to two old classes are randomly selected and connected (darker edges indicate more similar features). We detect distortion in the features of baseline (Figure 1(a)) relative to those of the target (Figure 1(c)), especially nodes located within the black dotted box. The two categories can be well separated in the feature distribution of target but not in the baseline model and lead to false predictions (grey nodes in Figure 1(a)). The topological correlation is also significantly changed, node #2 is in proximity to node #1 but is distant from node #3. Surprisingly, within the representation space of baseline, node #2 appears to be moving closer to node #3 while simultaneously becoming more distant from node #1, which exacerbates catastrophic forgetting. Our method (Figure 1(b)) designs graph information preservation modules to mitigate shifts successfully.

In this paper, we inspect GCIL from the perspective of information preservation and theoretically find a key factor in reducing catastrophic forgetting risk with hybrid methods is preserving old graph information. We correspond graph information into low-frequency local-global information and high-frequency information in spatial domain. Subsequently, a Graph Spatial Information Preservation (GSIP) framework is proposed for calibrating semantic and structural shifts. In detail, the old and new representations of nodes are obtained after replay graph data is input to old and new models. The old representations of a node are locally aligned with new representations of a node and its neighbors. Further, old and new representations are globally matched after mean pooling. Finally, new representations of nodes are encouraged to mimic neighbor distance similarities that appear in old representations.

The proposed GSIP can outperform existing replay designs by up to 10% in terms of the forgetting metric on large-scale datasets. It is easy to implement and can be easily adapted to information-preserving approaches to boost their performance. Experiments show that GSIP greatly improves

over current information-preserving methods under different experimental settings and calibrates node semantic and graph structure shifts. Our main contributions can be summarized as follows:

- We provide theoretical insights into GCIL and find that preserving old graph information corresponding to low-frequency local-global and high-frequency information in spatial domain can calibrate semantic and structural shifts and reduce catastrophic forgetting risk.
- We propose a simple yet effective method that utilizes node representations on old and new models to preserve node features, graph representations, and neighbor distances.
- By combining with graph replay-based methods, our framework consistently achieves performance improvements across several benchmark datasets and shows the effectiveness of all the proposed components.

## 2 Related Work

### 2.1 Incremental Learning

Incremental learning requires the model to retain the capability of predicting old tasks while acquiring information about new ones [3, 13, 14, 15, 16, 17, 18]. Class incremental learning is not assigned a task ID and has greater training difficulty than task incremental learning [19, 20]. Existing methods can be categorized into three groups. Parameter isolation methods dynamically adapt the model without restricting its structure and capacity, providing distinct parameters for each task [21, 22, 23, 24, 25]. Replay-based methods replay a subset of examples stored in previous tasks or generated using generative models to mitigate forgetting [26, 27, 28, 29]. Regularization-based methods introduce an additional regularization term in the loss function to prevent modifications to crucial parameters related to previous tasks [6, 30, 31, 32, 33].

Traditional incremental learning methods for images or text lack topology learning, making it challenging to achieve effective topology mining and information preservation. By contrast, we analyze the basics of preventing catastrophic forgetting in GCIL from information preservation and solve them in the spatial domain.

### 2.2 Graph Incremental Learning

Graph incremental learning focuses on handling streaming graph data, and numerous methods have been developed explicitly for graph data [34, 35, 36, 37, 38, 39, 40, 41]. Topology-aware Weight Preserving (TWP) preserves key parameters and topology of previous tasks through regularization terms [1]. Experience Replay Graph Neural Network (ERGNN) framework incorporates memory replay by storing representative nodes [7]. Sparsified Subgraph Memory (SSM) stores sampled sparse subgraphs in a memory repository to preserve structural information [8]. Su *et al.* introduced regularization terms to mitigate catastrophic forgetting from structural drift [11]. Zhang *et al.* redesigned the architecture into a three-layer prototype that adaptively selects different parameter combinations for different tasks [2]. The Condense and Train (CaT) [9] framework compresses the graph into a small but informative synthetic replay graph. Furthermore, two graph incremental learning benchmarks have recently been developed [42, 43].

In comparison, GSIP combines graph information preservation to avoid catastrophic forgetting through low-frequency local-global and high-frequency information preservation.

## 3 Problem Analysis

**Graph Class Incremental Learning (GCIL).** GCIL addresses the problem of supervised node classification within the context of an expanding graph. Specifically, each $G^t$ denotes a newly emerging subgraph within the overarching graph. A $G^t$ consists of a node set $V^t$ and an edge set $\mathcal{E}^t$ with its connectivity captured by adjacency matrix $A^t \in \mathbb{R}^{n \times n}$, where $n$ is the number of nodes. Each vertex $v$ is associated with node features $X_v$ and a target label $Y_v \in \{0, 1\}^c$, where $c$ represents the total number of classes. At time $t$, the GCIL problem denoted as $\mathbb{P}_{GCIL}$ is provided with a subgraph $G^t = \{X^t, A^t\}$. The $\mathbb{P}_{GCIL}$ problem is formally defined with the following signature:

$$\mathbb{P}_{GCIL}^t : \left\langle f^{t-1}, \left(G^t, Y^t\right), \mathcal{M}^{t-1}\right\rangle \to \left\langle f^t, \mathcal{M}^t\right\rangle, \tag{1}$$

where $f$ is graph neural networks and $\mathcal{M}$ signifies an external memory capable of storing a subset of training nodes or other useful graph data.

In the scenario of graph incremental learning, disparate tasks are mapped into distinct partitions of the graph. Once the learning for a specific task is completed, access to the corresponding data is restricted. Our objective is to learn a shared graph neural network model that distinguishes all classes from existing ones. Formally, we aim to minimize the loss caused by previously seen nodes at time step $\tau$ in $\mathbb{P}_{GCIL}$, the statistical risk of catastrophic forgetting is defined as:

$$\min_{\theta^\tau} \sum_{t=1}^{\tau} \mathbb{E}_{(G^t, Y^t)} \left[ \mathcal{H} \left( Y^t, \sigma \left( f \left( G^t; \theta^\tau \right) \right) \right) \right], \tag{2}$$

where $\theta$ indicates parameters of the model, $\mathcal{H}$ represents cross-entropy loss, and $\sigma$ denotes softmax activation function.

**Replay-Based GCIL.** Replay-based methods store replayed nodes or subgraphs in memory $\mathcal{M}$ by sampling. Catastrophic forgetting is solved by maintaining the historical distribution. These methods train a new model by minimizing loss of old task nodes on new model concerning the true labels. Given node representations on new model $Z^{new}(Z = f(\mathcal{M}; \theta))$, the replay loss is calculated as:

$$\mathcal{L}_{replay} = \frac{1}{|\mathcal{M}|} \sum_{i \in \mathcal{M}} \mathcal{H} \left( Y_i, \sigma \left( Z_i^{new} \right) \right). \tag{3}$$

**Semantic Shift and Structural Shift.** Due to memory and privacy limitations, a large amount of old graph data cannot be accessed in graph incremental learning, which leads to material information of old models being gradually forgotten and seriously damages new model performance on old classes. We design two novel shift metrics measuring semantic and structural forgetting degrees when trained on novel classes to show that model divergence manifests in node-level semantics and graph-level structure aspects. On CoraFull dataset, we conduct shift tests using model representations $Z^{old}$ and $Z^{new}$ generated by classical replay method ERGNN [7]. Specifically,

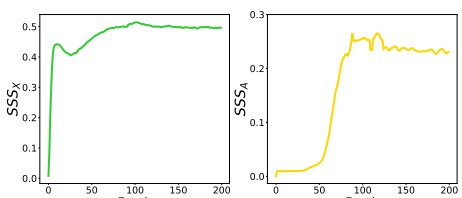

Figure 2: Semantic shift (left) and structural shift (right) between old and new models.

central kernel alignment [44] scheme is leveraged to compute Semantic Shift Score ($SSS_X$):

$$SSS_X(Z^{old}, Z^{new}) = 1 - \frac{HS(Z^{old}, Z^{new})}{\sqrt{HS(Z^{old}, Z^{old})HS(Z^{new}, Z^{new})}}, HS(Z^{old}, Z^{new}) = \frac{tr(Z^{old}CZ^{new}C)}{(n-1)^2}, \tag{4}$$

where $C$ is centering matrix $C_n = I_n - 1/n \mathbf{1}\mathbf{1}^\top$. In particular, Structural Shift Score ($SSS_A$) is derived by performing structure $\widehat{A}$ inference using feature cosine similarity, then computing differences between graph representations obtained by Anonymous Walk Embedding (AWE) [45]:

$$SSS_A(Z^{old}, Z^{new}) = 1 - COS(AWE(\widehat{A}^{old}), AWE(\widehat{A}^{new})), \widehat{A}_{ij} = \mathbb{1}\left[ COS(Z_i, Z_j) > \delta \right], \tag{5}$$

where cosine similarity function $COS(a, b) = a^\top b / (\|a\| \|b\|)$ is used to calculate feature similarity degree, and $\delta$ is similarity threshold. Each task is trained for 200 epochs, and shift scores range from 0 (no shift) to 1 (completely different). We observe that $SSS_X$ and $SSS_A$ in Figure 2 gradually rise with the increase of epochs. Serious shifts are found in both node semantic and graph structure between old and new models as new classes are trained.

## 4 Graph Spatial Information Preservation

### 4.1 Graph Information Preservation

Model information preservation for GCIL can be defined as the mutual information of graph information across old and new models when considering the corresponding model parameters:

$$\mathcal{P}_{\theta^{old} \to \theta^{new}} = \mathcal{I} \left( \mathscr{Z}^{old}; \mathscr{Z}^{new} \right), \tag{6}$$

here, $\mathscr{Z}^{old}$ and $\mathscr{Z}^{new}$ are graph information on old and new models. We directly maximize mutual information between $\mathscr{Z}^{old}$ and $\mathscr{Z}^{new}$, which inherits powerful encoding capability of $\theta^{old}$ to $\theta^{new}$.

**Proposition 1.** *The upper bound on graph information preservation can be estimated as:*

$$-\mathcal{I}\left(\mathscr{Z}^{old};\mathscr{Z}^{new}\right) \leq \left\|\mathscr{Z}^{old} - \mathscr{Z}^{new}\right\|_2^2 = \left\|\Delta\mathscr{Z}\right\|_2^2, \tag{7}$$

*we expect to maximize mutual information $\mathcal{I}\left(\mathscr{Z}^{old};\mathscr{Z}^{new}\right)$, thus minimizing $-\mathcal{I}\left(\mathscr{Z}^{old};\mathscr{Z}^{new}\right)$ in estimation is needed.*

Proposition 1 is proved in Appendix A.1, which suggests that graph information preservation is bounded with the square of Euclidean norm between old graph information $\mathscr{Z}^{old}$ and new graph information $\mathscr{Z}^{new}$.

## 4.2 Spatial Property

Based on the spatial properties of graphs, we analyze the maintenance of graph information in spatial domain to capture complex spatial relationships between nodes and edges in graphs.

**Lemma 1.** *(Graph spatial information factorization [46]) The graph convolution between convolution kernel $\mathcal{F}$ and the signal $\mathbf{x}$ to obtain graph information is formulated as follows:*

$$\mathcal{F} * \mathbf{x} = \frac{1}{2}\left(\mathcal{F}^l + \mathcal{F}^h\right) * \mathbf{x} = \frac{1}{2}\left(\mathcal{F}^l * \mathbf{x} + \mathcal{F}^h * \mathbf{x}\right) = \mathbf{x}, \tag{8}$$

*where $\mathcal{F}^l * \mathbf{x}$ and $\mathcal{F}^h * \mathbf{x}$ are low-/high- frequency graph information, $\mathcal{F}^l = I_n + \widetilde{D}^{-\frac{1}{2}}\widetilde{A}\widetilde{D}^{-\frac{1}{2}}$, $\mathcal{F}^h = I_n - \widetilde{D}^{-\frac{1}{2}}\widetilde{A}\widetilde{D}^{-\frac{1}{2}}$, $\widetilde{D}$ is diagonal degree matrix with $\widetilde{D}_{i,i} = \sum_j \widetilde{A}_{i,j}$, and $\widetilde{A} = A + I_n$ represents adjacency matrix with self-loop. Two pieces of information in spatial domain can be derived as follows:*

$$\mathcal{F}^l * \mathbf{x}_i \rightarrow \mathbf{x}_i^l = x_i + \sum_{j \in \mathcal{N}_i} \frac{x_j}{\sqrt{|\mathcal{N}_i||\mathcal{N}_j|}}, \mathcal{F}^h * \mathbf{x}_i \rightarrow \mathbf{x}_i^h = x_i - \sum_{j \in \mathcal{N}_i} \frac{x_j}{\sqrt{|\mathcal{N}_i||\mathcal{N}_j|}}, \tag{9}$$

*where $\mathcal{N}$ represents node neighbors.*

According to Lemma 1, there is an identity map that filters out graph information $\mathscr{Z}$ with graph convolution, which provides effective solutions to correspond $\mathscr{Z}^{old}$ and $\mathscr{Z}^{new}$ to spatial domain. For each component $i$, low-frequency information preserving $\left\|\Delta\mathscr{Z}_i^l\right\|_2^2$ is defined as:

$$\left\|\Delta\mathscr{Z}_i^l\right\|_2^2 = \left\|\left(Z_i^{old} + \sum_{j \in \mathcal{N}_i} \frac{Z_j^{old}}{\sqrt{|\mathcal{N}_i||\mathcal{N}_j|}}\right) - \left(Z_i^{new} + \sum_{j \in \mathcal{N}_i} \frac{Z_j^{new}}{\sqrt{|\mathcal{N}_i||\mathcal{N}_j|}}\right)\right\|_2^2, \tag{10}$$

where $Z^{old}$ and $Z^{new}$ denote obtained representations on old and new models. This implies that the low-frequency information preserving is a semantic gap between the sum of node features and their neighbor features of replay data on old and new models.

Sustained global connectivity is crucial to avert the erasure of global semantic information inherited from the preceding model. As displayed in Figure 3, global semantic shift does exist. We extend the concept of first-hop neighbor nodes in the previous equation to include the entire replay graph (*i.e.* multi-hop neighbors), which is denoted as:

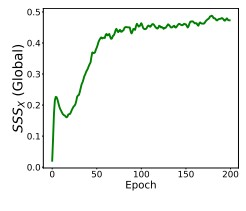

$$\left\|\Delta\mathscr{Z}_i^{\widehat{l}}\right\|_2^2 = \left\|\left(Z_i^{old} + \sum_{j \in \mathcal{M}} \frac{Z_j^{old}}{\sqrt{|\mathcal{M}||\mathcal{M}|}}\right) - \left(Z_i^{new} + \sum_{j \in \mathcal{M}} \frac{Z_j^{new}}{\sqrt{|\mathcal{M}||\mathcal{M}|}}\right)\right\|_2^2. \tag{11}$$

Similarly, the generalized low-frequency information preserving is the gap between the sum of node features and all replay data features on old and new models. It is worth noting that Eq. (10) provides a semantic comparison from a local perspective, whereas Eq. (11) compares from a global perspective.

Figure 3: Global semantic shift on old and new models.

For each component $i$, high-frequency information preserving $\left\|\Delta\mathscr{Z}_i^h\right\|_2^2$ is defined as:

$$\left\|\Delta\mathscr{Z}_i^h\right\|_2^2 = \left\|\left(Z_i^{old} - \sum_{j \in \mathcal{N}_i} \frac{Z_j^{old}}{\sqrt{|\mathcal{N}_i||\mathcal{N}_j|}}\right) - \left(Z_i^{new} - \sum_{j \in \mathcal{N}_i} \frac{Z_j^{new}}{\sqrt{|\mathcal{N}_i||\mathcal{N}_j|}}\right)\right\|_2^2. \tag{12}$$

High-frequency information preserving captures the gap between the difference in node features and neighbor features on old and new models from topological space.

Motivated by the above concepts, we introduce the following definition:

**Definition 1.** *(Graph spatial information preservation) A graph spatial information preservation model mainly consists of three kinds of information preservation $\left\|\Delta\mathscr{Z}\right\|_2^2 \approx \left\|\Delta\mathscr{Z}^l\right\|_2^2 \cup \left\|\Delta\mathscr{Z}^{\hat{l}}\right\|_2^2 \cup \left\|\Delta\mathscr{Z}^h\right\|_2^2$ (defined in Eq.* (10), (11), *and* (12)).

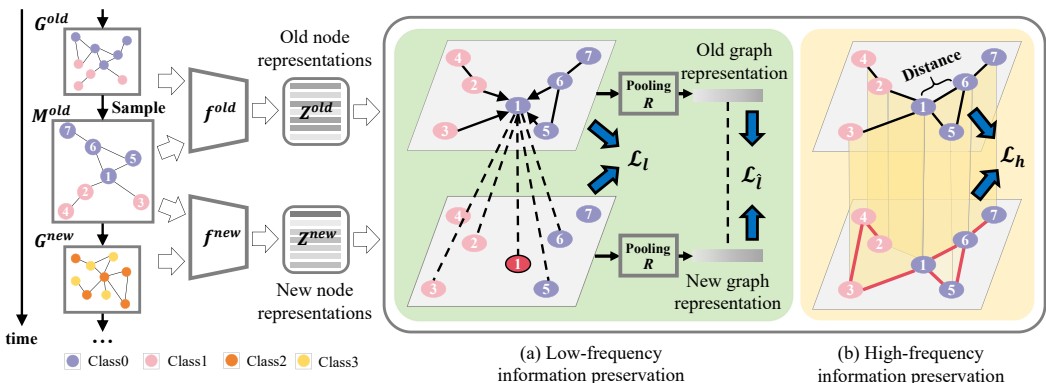

(a) Low-frequency information preservation

(b) High-frequency information preservation

Figure 4: A high-level overview of GSIP framework. It consists of low-/high- frequency modules to preserve old information. The old and new node representations are used to calculate information preserving loss of node representations, graph representations, and neighbor distances.

## 4.3 Instantiations for Graph Spatial Information Preservation

The above analysis yields two crucial insights: (1) old model information preservation can be solved by preserving the learned graph information; (2) graph information preserving can correspond to low-frequency local-global information and high-frequency information from spatial domain to calibrate node semantic and graph structure shifts. Inspired by these two insights, we propose low-/high-frequency information preservation to adequately capture the old model's information. A high-level overview of GSIP framework is shown in Figure 4. The pseudo-code can be found in Appendix A.4.

**Low-Frequency Information Preservation.** The node representations within the previous model are derived via iterative feature integration and neighborhood communication, so it contains low-frequency graph information. The information of old model is aligned into the neighborhood of new model to better utilize low-frequency information, which can be represented as:

$$\mathcal{L}_l = \sum_{i \in \mathcal{M}} \sum_{j \in \mathcal{N}_i \cup i} \left\| Z_i^{old}, Z_j^{new} \right\|_2^2, \tag{13}$$

where $Z$ is the output given by model. Low-frequency local information preserving loss uses Mean Squared Error (MSE) loss to locally match representations of nodes on old model $Z^{old}$ with representations of nodes and their neighbors on new model $Z^{new}$. For replay methods that do not explicitly save neighbors, neighbor selection can be found in Appendix A.2. It is worth noting that since inputs become sparse when converted to probabilities, the softmax followed by Kullback Leibler (KL) divergence loss is not applied [47].

Preserving global information about low-frequency components aligns old model information as a whole and prevents catastrophic forgetting. Low-frequency global information preserving loss is introduced to minimize difference between global representations of old and new models, which is defined as:

$$\mathcal{L}_{\hat{l}} = \left\| R(Z^{old}), R(Z^{new}) \right\|_2^2, \tag{14}$$

where $R$ represents pooling method, which is computed by mean pooling $R(Z) = 1/|\mathcal{M}| \sum_{i \in \mathcal{M}} Z_i$. Similarly, MSE loss is used to calculate global representation gaps.

**High-Frequency Information Preservation.** The high-frequency part of spatial domain represents the difference between the features of nodes and neighbors. The updated model preserves old topology by incorporating prior local contextual information and then mitigates heterogeneous information propagation blockage caused by smoothness assumption. Specifically, for node $v_i$, $\mathcal{N}_i$ denotes neighborhood node set and defines $S^{old}(v_i, \mathcal{N}_i)$ as the similarity of selected node vector with adjacent nodes computed by old model:

$$S^{old}(v_i, \mathcal{N}_i) = \left[ S_1^{old}, \ldots, S_{|\mathcal{N}_i|}^{old} \right], S_j^{old} = \frac{\exp\left( \mathcal{K}\left( Z_i^{old}, Z_j^{old} \right) \right)}{\sum_{j' \in \mathcal{N}_i} \exp\left( \mathcal{K}\left( Z_i^{old}, Z_{j'}^{old} \right) \right)}, \tag{15}$$

where $\mathcal{K}(\cdot, \cdot)$ represents kernel function that measures pairwise distances between each node and its neighbors in the latent feature space, and element-wise absolute values $\mathcal{K}(Z_i, Z_j) = |Z_i - Z_j|$ is used. Then, we measure similarity distribution from new model $S^{new}(v_i, \mathcal{N}_i)$, which is formed by:

$$S^{new}(v_i, \mathcal{N}_i) = \left[ S_1^{new}, \ldots, S_{|\mathcal{N}_i|}^{new} \right], S_j^{new} = \frac{\exp\left( \mathcal{K}\left( Z_i^{new}, Z_j^{new} \right) \right)}{\sum_{j' \in \mathcal{N}_i} \exp\left( \mathcal{K}\left( Z_i^{new}, Z_{j'}^{new} \right) \right)}. \tag{16}$$

High-frequency information preserving is proposed to map neighborhood pairwise differences between old and new models in topological space, information loss from old structure to new structure is more easily recognized with the help of KL divergence, which is denoted as follows:

$$\mathcal{L}_h = \sum_{i \in \mathcal{M}} S^{old}(v_i, \mathcal{N}_i) \log \frac{S^{old}(v_i, \mathcal{N}_i)}{S^{new}(v_i, \mathcal{N}_i)}. \tag{17}$$

**Model Learning.** To combine different preserving losses, the final graph information preservation loss function is defined as:

$$\mathcal{L}_{gip} = \mathcal{L}_l + \beta \mathcal{L}_{\hat{l}} + \gamma \mathcal{L}_h, \tag{18}$$

where $\beta$ and $\gamma$ are loss weights.

Node classification loss is obtained by $\mathcal{L}_{nc} = 1/|G| \sum_{i \in G} \mathcal{H}(Y_i, \sigma(f(G; \theta^{new})))$. Therefore, the overall model learning objective is the weighted sum of current node classification loss, replay loss, and graph information preserving loss:

$$\mathcal{L} = \mathcal{L}_{nc} + \alpha_{replay} \mathcal{L}_{replay} + \alpha_{gip} \mathcal{L}_{gip}, \tag{19}$$

where $\alpha_{replay}$ and $\alpha_{gip}$ are loss weights, and the value of $\alpha_{replay}$ is relevant to the design of replayed method. More analysis about the preservation of other graph frequency information (*i.e.* mid-frequency information and high-frequency global information) is given in Appendix A.3.

## 5 Experiments

### 5.1 Datasets and Setups

**Datasets and Settings.** We utilize five public datasets to evaluate the effectiveness of the proposed method in GCIL, the statistics of datasets are reported in Appendix B.1. Three ways of dividing classes are used: one is divided unequally, and the other two are divided equally, with equal classes per task. The first dataset is CoraFull [48], which has 70 classes, 30 classes are used as base classes for dividing unequally, then 20 classes are used as an increment, and we divide classes equally into 10 or 2 classes per task. Arxiv [49] and Reddit [50], both containing 40 classes, dividing unequally using 10 classes as base classes, then in increments of 5 classes, and dividing equally with 10 or 2 classes per task. Each dataset has 3 tasks with 2 classes per task on Cora [51] and Citeseer [51]. The latest benchmark [42] is employed to implement ERGNN, along with CaT [9] is used to implement SSM and CaT, we follow their settings in graph class incremental learning. Our implementation and detailed settings are available in Appendix B.4 and B.5.

**Baselines and Metrics.** We compare our method with the following baselines, including Finetuning, Joint, EWC [30], GEM [31], MAS [32], LwF [6], TWP [1], SSRM [11], and three replay-based methods (*i.e.* ERGNN [7], SSM [8], and CaT [9]), where three graph replay methods apply our framework. Finetuning is the lower bound baseline updating the model only with newly emerging graph data. Joint is the ideal upper bound and inputs contain all previous graph data. We choose two widely used metrics to evaluate the performance of the compared methods, including Average Performance (AP) and Average Forgetting (AF) [31].

Table 1: Performance comparison on CoraFull, Arxiv, and Reddit for GCIL setting. Results are averaged among three trials. The best performing results (excluding Joint) are highlighted in **bold**.

| Method | CoraFull | | | | | | Arxiv | | | | | | Reddit | | | | | |
|---|---|---|---|---|---|---|---|---|---|---|---|---|---|---|---|---|---|---|
| | Unequally | | Equally (10) | | Equally (2) | | Unequally | | Equally (10) | | Equally (2) | | Unequally | | Equally (10) | | Equally (2) | |
| | AP↑ | AF↑ | AP↑ | AF↑ | AP↑ | AF↑ | AP↑ | AF↑ | AP↑ | AF↑ | AP↑ | AF↑ | AP↑ | AF↑ | AP↑ | AF↑ | AP↑ | AF↑ |
| Finetuning | 23.95 | -76.59 | 11.06 | -85.77 | 2.70 | -95.48 | 11.64 | -70.41 | 5.41 | -50.00 | 4.91 | -87.61 | 14.66 | -91.80 | 22.90 | -94.42 | 5.83 | -94.23 |
| EWC | 24.09 | -75.78 | 11.15 | -86.08 | 5.13 | -93.08 | 11.93 | -68.97 | 14.83 | -57.33 | 4.91 | -87.58 | 13.79 | -95.35 | 22.30 | -95.27 | 9.66 | -93.85 |
| GEM | 23.95 | -76.05 | 11.23 | -85.78 | 7.97 | -90.00 | 11.61 | -60.27 | 8.27 | -44.42 | 4.92 | -86.66 | 18.51 | -89.79 | 22.58 | -93.93 | 35.11 | -65.67 |
| MAS | 24.20 | -75.97 | 10.94 | -82.37 | 4.43 | -89.22 | 11.09 | -66.76 | 12.32 | -57.99 | 5.29 | -81.64 | 15.45 | -0.50 | 25.54 | 0.01 | 5.98 | -14.17 |
| LwF | 23.99 | -76.14 | 11.14 | -85.67 | 2.72 | -95.08 | 11.93 | -70.66 | 14.69 | -58.93 | 4.91 | -88.14 | 16.13 | -90.31 | 24.39 | -93.29 | 7.59 | -88.98 |
| TWP | 23.86 | -75.74 | 11.01 | -85.43 | 3.56 | -94.66 | 11.93 | -69.26 | 14.41 | -56.56 | 4.90 | -87.75 | 13.95 | -96.17 | 21.22 | -96.41 | 9.34 | -94.24 |
| SSRM | 63.62 | -16.24 | 31.39 | -60.61 | 3.22 | -89.29 | 31.51 | -45.12 | 26.61 | -46.22 | 26.16 | -61.24 | 78.40 | -20.92 | 76.78 | -23.16 | 83.96 | -15.41 |
| ERGNN | 60.91 | -19.47 | 24.39 | -69.31 | 3.01 | -94.34 | 31.18 | -45.45 | 24.47 | -49.11 | 24.70 | -62.26 | 76.60 | -23.22 | 75.22 | -25.26 | 83.16 | -16.21 |
| **+GSIP** | 67.22 | -10.91 | 71.15 | -11.37 | 44.79 | -44.60 | 34.09 | -32.59 | 33.88 | -27.97 | 40.21 | -28.96 | 90.82 | -6.05 | 89.59 | -2.03 | 93.03 | -5.50 |
| **Improve** ↑ | 6.31 | 8.56 | 46.76 | 57.94 | 41.78 | 49.74 | 2.91 | 12.86 | 9.41 | 21.14 | 15.51 | 33.30 | 14.22 | 17.17 | 14.37 | 23.23 | 9.87 | 10.71 |
| SSM | 50.51 | -10.56 | 62.90 | -6.02 | 79.02 | -4.24 | 63.48 | -12.41 | 60.57 | -10.09 | 63.91 | -12.48 | 90.10 | -5.83 | 86.91 | -3.24 | 96.24 | -1.64 |
| **+GSIP** | 55.32 | -2.50 | 63.86 | **0.08** | 79.31 | 0.70 | 63.36 | -7.27 | 61.34 | -6.34 | 64.16 | -8.87 | 90.74 | -3.97 | 87.41 | 0.13 | 96.25 | -0.65 |
| **Improve** ↑ | 4.81 | 8.06 | 0.96 | 6.10 | 0.29 | 4.94 | -0.12 | 5.14 | 0.77 | 3.75 | 0.25 | 3.61 | 0.64 | 1.86 | 0.50 | 3.37 | 0.01 | 0.99 |
| CaT | 70.55 | -5.26 | 76.35 | -5.44 | 80.64 | -4.31 | **71.66** | -8.33 | 70.16 | -7.25 | 66.21 | -12.73 | **96.39** | -0.77 | 93.97 | -1.31 | **97.64** | -0.49 |
| **+GSIP** | **71.06** | **-0.28** | **78.29** | -1.25 | **81.10** | **2.68** | 71.52 | **-4.76** | **70.57** | **-3.97** | **68.80** | **3.49** | 96.15 | **-0.23** | **94.23** | **0.21** | 97.55 | **1.04** |
| **Improve** ↑ | 0.51 | 4.98 | 1.94 | 4.19 | 0.46 | 6.99 | -0.14 | 3.57 | 0.41 | 3.28 | 2.59 | 16.22 | -0.24 | 0.54 | 0.26 | 1.52 | -0.09 | 1.53 |
| Joint | 85.3 | - | 85.3 | - | 85.3 | - | 63.5 | - | 63.5 | - | 63.5 | - | 98.2 | - | 98.2 | - | 98.2 | - |

## 5.2 Performance Comparison

**GSIP can improve the performance of existing replay-based information preservation methods.** The effect of GCIL on five datasets is presented in Table 1 and Table 2, and the results with standard deviation are presented in Appendix C.1. Joint does not provide AF due to its non-compliance with the incremental learning setting. The existing regularization term relies on the correlation between old and new classes leading to catastrophic forgetting, and some of them do not take topology into account. The hybrid method SSRM absorbs partial old information, resulting in extremely limited performance gains. GSIP consistently demonstrates significant improvements in AP and AF when combined with existing replay methods. For example, ERGNN-GSIP improves both AP and AF by about 10% on Reddit. On CoraFull, SSM-GSIP under unequal partitioning situation improves AP and AF by 4.81% and 8.06%, respectively. CaT-GSIP performs remarkably well on Arxiv, even surpassing the performance of Joint, which has a 16.22% increase in AF on a setting with an increment of 2. CaT experiences a slight decrease in AP in some settings after using GSIP. This can be

Table 2: Performance comparison on Cora and Citeseer for GCIL setting. Results are averaged among three trials. The best performing results (excluding Joint) are highlighted in **bold**.

| Method | Cora | | Citeseer | |
|---|---|---|---|---|
| | Equally (2) | | Equally (2) | |
| | AP↑ | AF↑ | AP↑ | AF↑ |
| Finetuning | 32.58 | -96.83 | 31.46 | -77.86 |
| EWC | 32.58 | -97.16 | 31.26 | -78.22 |
| GEM | 32.70 | -97.12 | 31.39 | -77.70 |
| MAS | 31.84 | -97.17 | 31.25 | -76.67 |
| LwF | 32.58 | -97.57 | 31.44 | -78.29 |
| TWP | 32.58 | -97.32 | 31.22 | -78.14 |
| SSRM | 35.48 | -70.01 | 51.91 | -67.66 |
| ERGNN | 65.48 | -46.09 | 47.65 | -51.12 |
| **+GSIP** | 71.29 | -36.95 | 61.29 | -29.38 |
| **Improve** ↑ | 5.81 | 9.14 | 13.64 | 21.74 |
| SSM | 67.64 | -19.78 | 60.99 | -13.60 |
| **+GSIP** | 69.92 | -11.82 | 61.86 | **-8.39** |
| **Improve** ↑ | 2.28 | 7.96 | 0.87 | 5.21 |
| CaT | 88.22 | -4.40 | 75.08 | -10.93 |
| **+GSIP** | **89.60** | **1.84** | **77.02** | -9.95 |
| **Improve** ↑ | 1.38 | 6.24 | 1.94 | 0.98 |
| Joint | 93.09 | - | 78.27 | - |

attributed to GSIP enabling better preservation of graph spatial information from old model, resulting in a lower forgetting rate. In addition, for ERGNN-GSIP, the AP and AF increase by 5.81% and 9.14% on Cora and by 13.64% and 21.74% on Citeseer. On Cora and Citeseer datasets, the AF of SSM-GSIP improves by more than 5%, and CaT-GSIP achieves the highest performance in most cases, with the AP approaching the value of Joint.

**GSIP can consistently achieve excellent performance on old classes in different task IDs.** The performance matrices of ERGNN on CoraFull before and after incorporating GSIP are shown in Figure 5(a) and Figure 5(b). It is difficult to remember the information of old classes before employing graph spatial information preservation. After implementing the GSIP scheme, the performance matrix demonstrates a deceleration in the forgetting process (*i.e.*, the color of each column does not change much and the color deepens), which indicates that the catastrophic forgetting problem is mitigated due to the preserving of old graph information.

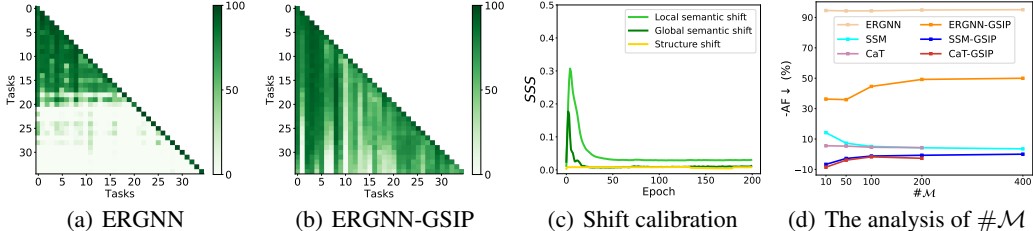

|  (a) ERGNN | (b) ERGNN-GSIP | (c) Shift calibration | (d) The analysis of $\#\mathcal{M}$ |

Figure 5: (a)-(b) Performance matrices on CoraFull dataset. (c) Semantic and structural shift calibration of old and new models during increments. (d) Performance changes affected by $\#\mathcal{M}$ on CoraFull dataset.

**GSIP can offer information preservation capability to calibrate semantic shift and structural shift.** Figure 5(c) exhibits the curves of shift scores during the incremental process for ERGNN on CoraFull. It can be noted that shift scores start at a relatively high value, gradually decrease, and smooth out after graph spatial information is maintained, demonstrating that the low-frequency local-global information and high-frequency information of old model are well captured, and semantic and structural shifts are nicely calibrated.

## 5.3 Ablation Study

We investigate the effectiveness of low-frequency local modules (LL), low-frequency global modules (LG), and high-frequency modules (H), the experimental results use ERGNN as baseline (B). The results of ERGNN, SSM, and CaT with standard deviation are summarized in Appendix C.2. The above components are added one by one to baseline for performance comparison. From Table 3 we observe that: (1) When LL is utilized, the model can easily learn the aggregation rules of nodes and neighbors from old model locally. AP (AF) improves by about 2% (6%) to 45% (55%) over the baseline demonstrating the superiority of LL. (2) Combining LG significantly improves performance, especially on Reddit. The reason may be that larger datasets have greater overall shifts during increments. (3) H also brings significant improvements, with AP (AF) improving by about 1.6% (0.9%) to 2.4% (7.4%) over B+LL+LG on Reddit. This indicates that the H module can extract more topology information for better performance.

Table 3: Ablation comparisons of graph spatial information preservation.

| Method | CoraFull | | | | | | Arxiv | | | | | | Reddit | | | | | |
| | Unequally | | Equally (10) | | Equally (2) | | Unequally | | Equally (10) | | Equally (2) | | Unequally | | Equally (10) | | Equally (2) | |
| | AP↑ | AF↑ | AP↑ | AF↑ | AP↑ | AF↑ | AP↑ | AF↑ | AP↑ | AF↑ | AP↑ | AF↑ | AP↑ | AF↑ | AP↑ | AF↑ | AP↑ | AF↑ |
| B | 60.91 | -19.47 | 24.39 | -69.31 | 3.01 | -94.34 | 31.18 | -45.45 | 24.47 | -49.11 | 24.70 | -62.26 | 76.60 | -23.22 | 75.22 | -25.26 | 83.16 | -16.21 |
| B+LL | 65.79 | -13.20 | 69.02 | -14.15 | 41.37 | -47.39 | 33.27 | -34.60 | 27.10 | -40.95 | 38.09 | -35.04 | 84.63 | -12.09 | 84.26 | -12.37 | 87.52 | -10.97 |
| B+LL+LG | 66.22 | -12.78 | 69.77 | -13.13 | 41.84 | -46.73 | 34.00 | -33.61 | 32.80 | -34.40 | 39.89 | **-28.85** | 89.21 | -6.97 | 87.17 | -9.45 | 91.34 | -7.12 |
| B+LL+LG+H | **67.22** | **-10.91** | **71.15** | **-11.37** | **44.79** | **-44.60** | **34.09** | **-32.59** | **33.88** | **-27.97** | **40.21** | -28.96 | **90.82** | **-6.05** | **89.59** | **-2.03** | **93.03** | **-5.50** |

## 5.4 Further Analysis

**Hyper-Parameter Analysis.** We analyze the impact of the number of storage nodes for each task $\#\mathcal{M}$ on performance. As depicted in Figure 5(d), it can be observed that the proposed method consistently outperforms the original method in terms of the -AF metric (the lower, the better), regardless of the value of $\#\mathcal{M}$. Interestingly, even with less memory, the proposed method still achieves better performance. CaT cannot be trained on 400 nodes due to Cuda memory constraints. We analyze the impact of loss weight $\alpha_{gip}$ on ERGNN, SSM, and CaT across CoraFull, Arxiv, and Reddit datasets with increments of 2 in Figure 6. For ERGNN, SSM, and CaT, $\alpha_{gip,1}$ is set to $[1, 1, 0.1]$, $[0.01, 0.01, 0.01]$, and $[0.1, 0.01, 0.01]$ for three datasets. It can be observed that the performance change is not as significant with the variation of $\alpha_{gip}$ on SSM and CaT. However, different $\alpha_{gip}$ has a greater impact on performance with ERGNN-GSIP. The possible reason is that ERGNN selects representative nodes for replay, which may cause class imbalance and topology discarding. For ERGNN, SSM, and CaT, the optimal hyper-parameters $\alpha_{gip}$ on three datasets are $[50, 10, 1]$, $[0.1, 0.1, 0.1]$, and $[1, 0.5, 0.5]$. Because of space limitation, we provide more curves about $\#\mathcal{M}$ in Appendix C.3 and hyper-parameter analysis of loss weights $\beta$ and $\gamma$ in Appendix C.4.

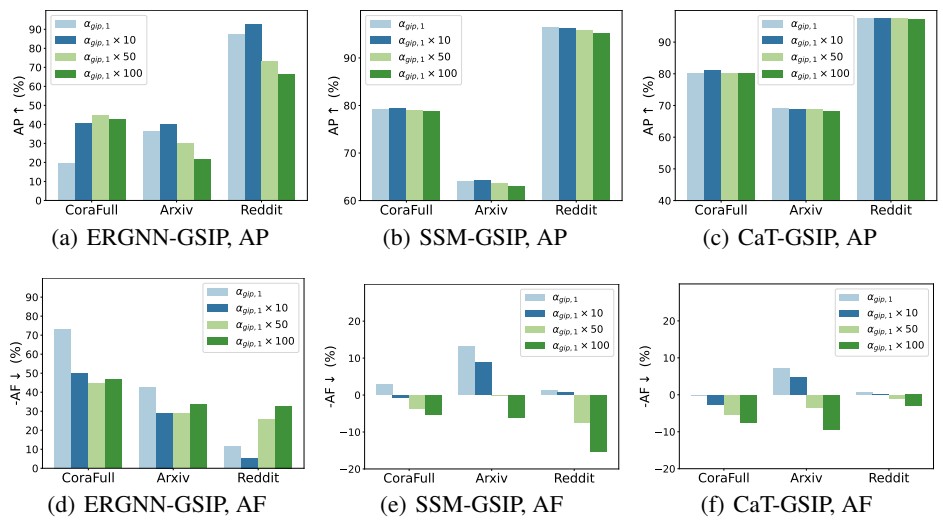

Figure 6: The analysis of $\alpha_{gip}$ in ERGNN, SSM, and CaT on CoraFull, Arxiv, and Reddit datasets.

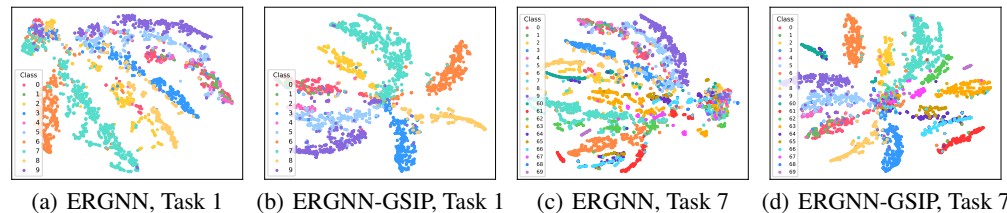

(a) ERGNN, Task 1    (b) ERGNN-GSIP, Task 1    (c) ERGNN, Task 7    (d) ERGNN-GSIP, Task 7

Figure 7: The visualization of node embeddings in Task 1 and Task 7 on CoraFull dataset.

**Visulization.** To qualitatively demonstrate the effectiveness of our representations, we adopt t-SNE [52] to visualize the learned node embeddings. After learning the last task, Figure 7(a) and Figure 7(b) show the results of the learned node embeddings in Task 1 on CoraFull, while Figure 7(c) and Figure 7(d) demonstrate the results of the last task. We can clearly observe that GSIP possesses better representation ability by considering representations and classifying old and new classes well.

## 6   Conclusion

We contribute to the literature of GCIL by addressing the issue of information preservation from old model when adapting to new classes. The key insight is that preserving graph information from spatial domain plays a vital role in preserving information about old model, and subsequently calibrates semantic and structural shifts and reduces catastrophic forgetting risk. To accomplish this objective, we introduce a framework, GSIP, which utilizes the outputs of nodes in old model to diffuse the outputs of new model and its neighbors, then aligns the outputs of new model with old model after pooling. Finally, GSIP maintains kernel distance of neighbor pairs on both old and new models. The graph information is remembered by low-frequency local-global information preserving and high-frequency information preserving in feature and topological space. Evaluations over benchmark datasets show the superiority of GSIP in handling different dataset splitting cases. In the future, we will investigate comprehensive analysis for the preservation of complicated graph signals.

## Acknowledgement

This work was supported in part by the National Science and Technology Major Project under Grant 2022ZD0116500, in part by the National Natural Science Foundation of China under Grants 62436002, 62476195, 61925602, U23B2049, and 62222608, in part by Tianjin Natural Science Funds for Distinguished Young Scholar under Grant 23JCJQJC00270, and in part by Zhejiang Provincial Natural Science Foundation of China under Grant LD24F020004.

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

## A Method

### A.1 The Proof of Proposition 1

**Proposition 1.** *The upper bound on graph information preservation can be estimated as:*

$$-\mathcal{I}\left(\mathscr{Z}^{old};\mathscr{Z}^{new}\right) \leq \|\mathscr{Z}^{old} - \mathscr{Z}^{new}\|_2^2 = \|\Delta\mathscr{Z}\|_2^2, \tag{20}$$

*we expect to maximize mutual information $\mathcal{I}\left(\mathscr{Z}^{old};\mathscr{Z}^{new}\right)$, thus minimizing $-\mathcal{I}\left(\mathscr{Z}^{old};\mathscr{Z}^{new}\right)$ in estimation is needed.*

*Proof.* The mutual information of graph information on old model and new model [53] can be equivalent to:

$$
\begin{aligned}
\mathcal{I}\left(\mathscr{Z}^{old};\mathscr{Z}^{new}\right) &= \int d\mathscr{Z}^{new} d\mathscr{Z}^{old} p(\mathscr{Z}^{new},\mathscr{Z}^{old}) \log\left(\frac{p(\mathscr{Z}^{new},\mathscr{Z}^{old})}{p(\mathscr{Z}^{new})p(\mathscr{Z}^{old})}\right) \\
&= \sum_{\mathscr{Z}^{new},\mathscr{Z}^{old}} p(\mathscr{Z}^{new},\mathscr{Z}^{old}) \log\left(\frac{p(\mathscr{Z}^{new},\mathscr{Z}^{old})}{p(\mathscr{Z}^{new})p(\mathscr{Z}^{old})}\right) \\
&= \sum_{\mathscr{Z}^{new},\mathscr{Z}^{old}} p(\mathscr{Z}^{new},\mathscr{Z}^{old}) \log\left(\frac{p(\mathscr{Z}^{new},\mathscr{Z}^{old})}{p(\mathscr{Z}^{new})}\right) \\
&\quad - \sum_{\mathscr{Z}^{new},\mathscr{Z}^{old}} p(\mathscr{Z}^{new},\mathscr{Z}^{old}) \log p(\mathscr{Z}^{old}) \\
&= \sum_{\mathscr{Z}^{new},\mathscr{Z}^{old}} p(\mathscr{Z}^{new}) p(\mathscr{Z}^{old}\mid\mathscr{Z}^{new}) \log(p(\mathscr{Z}^{old}\mid\mathscr{Z}^{new})) \\
&\quad - \sum_{\mathscr{Z}^{new},\mathscr{Z}^{old}} p(\mathscr{Z}^{new},\mathscr{Z}^{old}) \log p(\mathscr{Z}^{old}) \\
&= -\sum_{\mathscr{Z}^{new}} p(\mathscr{Z}^{new}) \mathcal{H}(\mathscr{Z}^{old}\mid\mathscr{Z}^{new}=\mathscr{Z}^{new}) - \sum_{\mathscr{Z}^{old}} \log p(\mathscr{Z}^{old}) p(\mathscr{Z}^{old}) \\
&= \mathcal{H}\left(\mathscr{Z}^{old}\right) - \mathcal{H}\left(\mathscr{Z}^{old}\mid\mathscr{Z}^{new}\right) \\
&= \mathcal{H}(\mathscr{Z}^{old}) + \mathbb{E}_{\mathscr{Z}^{old},\mathscr{Z}^{new}}[\log p(\mathscr{Z}^{old}\mid\mathscr{Z}^{new})] \\
&= \mathcal{H}(\mathscr{Z}^{old}) + \mathbb{E}_{\mathscr{Z}^{old},\mathscr{Z}^{new}}[\log q(\mathscr{Z}^{old}\mid\mathscr{Z}^{new})] \\
&\quad + \mathbb{E}_{\mathscr{Z}^{new}}\left[\mathcal{D}_{KL}(p(\mathscr{Z}^{old}\mid\mathscr{Z}^{new})\|q(\mathscr{Z}^{old}\mid\mathscr{Z}^{new}))\right] \\
&\geq \mathbb{E}_{\mathscr{Z}^{old},\mathscr{Z}^{new}}[\log q(\mathscr{Z}^{old}\mid\mathscr{Z}^{new})],
\end{aligned}
\tag{21}
$$

where expectations are over distribution $p(\mathscr{Z}^{old},\mathscr{Z}^{new})$, $\mathcal{H}(\mathscr{Z}^{old})$ has been removed since it is constant with respect to optimized parameters. The reason for the inequality is the non-negativity of entropy $\mathcal{H}$ and Kullback-Leiber divergence $\mathcal{D}_{KL}$.

We expect to maximize mutual information $\mathcal{I}\left(\mathscr{Z}^{old};\mathscr{Z}^{new}\right)$, so we need to minimize $-\mathcal{I}\left(\mathscr{Z}^{old};\mathscr{Z}^{new}\right)$ in the loss function, hence we can obtain:

$$-\mathcal{I}\left(\mathscr{Z}^{old};\mathscr{Z}^{new}\right) \leq -\mathbb{E}_{\mathscr{Z}^{old},\mathscr{Z}^{new}}[\log q(\mathscr{Z}^{old}\mid\mathscr{Z}^{new})]. \tag{22}$$

The conditional likelihood is maximized to fit the information of old model, and the new model receives compressed information needed to recover old model. A Gaussian distribution with mean $\boldsymbol{\mu}$ and variance $\boldsymbol{\sigma}$ is employed to estimate variational distribution $q(\mathscr{Z}^{old}\mid\mathscr{Z}^{new})$, which is expressed as follows:

$$
\begin{aligned}
-\mathcal{I}\left(\mathscr{Z}^{old};\mathscr{Z}^{new}\right) &\leq -\sum_{i=1}^{|\mathcal{M}|} \log q\left(\mathscr{Z}_i^{old}\mid\mathscr{Z}^{new}\right) \\
&= \sum_{i=1}^{|\mathcal{M}|} \log \boldsymbol{\sigma}_i + \frac{\left(\mathscr{Z}_i^{old} - \boldsymbol{\mu}_i(\mathscr{Z}^{new})\right)^2}{2\boldsymbol{\sigma}_i^2} + \mathcal{C},
\end{aligned}
\tag{23}
$$

where $\mathcal{C}$ is constant. Mean squared error matching can be seen as a specific instance of the above formula when $\boldsymbol{\mu}(\mathscr{Z}^{new}) = \mathscr{Z}^{new}$ and $\boldsymbol{\sigma} = 1$, thus $-\mathcal{I}\left(\mathscr{Z}^{old};\mathscr{Z}^{new}\right) \leq \|\mathscr{Z}^{old} - \mathscr{Z}^{new}\|_2^2$. $\qquad\square$

## A.2 Neighbor Selection

Some replay-based methods do not explicitly save topology, so we treat nodes with high representation similarity and label correlation on old model as neighbors for each node when computing low-frequency local information preserving loss. Nodes whose feature similarity is greater than the given threshold and whose labels are the same can be neighbors, formulated as follows:

$$\mathcal{N}_i = \{j \mid (COS(Z_i^{old}, Z_j^{old}) > \widehat{\delta}) \cap (Y_i = Y_j), \forall j \in \mathcal{M}\}, i \in \mathcal{M}, \tag{24}$$

where the cosine similarity function $COS(Z_i, Z_j) = Z_i^\top Z_j / (\|Z_i\| \|Z_j\|)$ is used to calculate the degree of feature similarity, and $\widehat{\delta}$ is similarity threshold.

We only use node feature similarity when calculating the loss of high-frequency information preservation for neighbor selection. The absence of label correlation captures the distance between nodes on different classes, which is conducive to the model to distinguish different classes. The neighborhood selection process is expressed as follows:

$$\mathcal{N}_i = \{j \mid COS(Z_i^{old}, Z_j^{old}) > \delta, \forall j \in \mathcal{M}\}, i \in \mathcal{M}, \tag{25}$$

where $\delta$ is the similarity threshold.

## A.3 The Analysis of Other Graph Frequency Information Preservation

We perform analyses and experiments to assess the preservation of various aspects of graph frequency information, including mid-frequency information and high-frequency global information.

**Mid-Frequency Information Preservation.** According to the definition of mid-frequency filtering graph convolutional networks [54, 55], the mid-frequency convolution can be expressed as:

$$\mathcal{F}^m = (I_n - \widetilde{D}^{-\frac{1}{2}} \widetilde{A} \widetilde{D}^{-\frac{1}{2}})(I_n + \widetilde{D}^{-\frac{1}{2}} \widetilde{A} \widetilde{D}^{-\frac{1}{2}}). \tag{26}$$

Through a similar analysis as mentioned above, mid-frequency information preservation is defined as:

$$\|\Delta \mathscr{Z}_i^m\|_2^2 = \left\| \left( Z_i^{old} - \sum_{j \in \mathcal{N}_i^2} \frac{Z_j^{old}}{\sqrt{|\mathcal{N}_i^2| |\mathcal{N}_j^2|}} \right) - \left( Z_i^{new} - \sum_{j \in \mathcal{N}_i^2} \frac{Z_j^{new}}{\sqrt{|\mathcal{N}_i^2| |\mathcal{N}_j^2|}} \right) \right\|_2^2, \tag{27}$$

where $\mathcal{N}^2$ is the second-hop neighbors of nodes. The distinction between mid-/high- frequency information preservation lies in that mid-frequency signals calculate the differences between the node and its second-hop neighbors.

**High-Frequency Global Information Preservation.** The general formula for high-frequency global information preservation is represented as follows:

$$\left\| \Delta \mathscr{Z}_i^{\widehat{h}} \right\|_2^2 = \left\| \left( Z_i^{old} - \sum_{j \in \mathcal{M}} \frac{Z_j^{old}}{\sqrt{|\mathcal{M}| |\mathcal{M}|}} \right) - \left( Z_i^{new} - \sum_{j \in \mathcal{M}} \frac{Z_j^{new}}{\sqrt{|\mathcal{M}| |\mathcal{M}|}} \right) \right\|_2^2. \tag{28}$$

It implies that every node in replayed graph needs to calculate disparity with other nodes. Since graph neural network assumes that neighboring nodes have similar representations, the penalizing distance between nodes and their neighbors at multiple hops away is redundant and does not contribute to structural preservation. Moreover, the inclusion of this term imposes an optimization burden and exhibits high time complexity ($O(|\mathcal{M}|^2 \cdot \Bbbk)$, where $\Bbbk$ denotes the dimensions of the hidden spaces).

We add mid-frequency information preservation (**M**) and high-frequency global information preservation (**HG**) to GSIP in Table 4, which can yield a slight performance improvement in some cases. However, it does not lead to better performance enhancements or outstanding results. The possible reason is that the preservation of first-hop neighbors is sufficient to calibrate structural shift.

## A.4 Algorithm

The proposed method is summarized in Algorithm 1.

Table 4: Performance comparison before and after adding other graph frequency information preservation on CoraFull dataset.

| Method | Unequally | | | | Equally (10) | | | | Equally (2) | | | |
|---|---|---|---|---|---|---|---|---|---|---|---|---|
| | AP↑ | AF↑ | AP↑ | AF↑ | AP↑ | AF↑ | AP↑ | AF↑ | AP↑ | AF↑ | AP↑ | AF↑ |
| GSIP | **55.32** | **-2.50** | **67.22** | -10.91 | 63.86 | 0.08 | **71.15** | **-11.37** | **79.31** | 0.70 | **44.79** | **-44.60** |
| | ±0.75 | ±1.13 | ±0.44 | ±0.62 | ±0.85 | ±0.76 | ±0.98 | ±0.74 | ±0.50 | ±0.25 | ±1.77 | ±1.67 |
| +M / HG | 55.28 | -2.61 | 66.63 | **-10.07** | **63.89** | **0.20** | 69.96 | -12.85 | 79.29 | **0.75** | 24.66 | -68.48 |
| | ±0.65 | ±1.04 | ±0.49 | ±1.75 | ±0.86 | ±0.68 | ±0.85 | ±0.94 | ±0.67 | ±0.52 | ±1.47 | ±1.35 |

---

**Algorithm 1** Framework of GSIP

---

**Input:** At time step $t > 1$: New input $G$, Memory $\mathcal{M}$, Graph neural networks $f$, Labels $Y$, Learned parameter $\theta^{old}$, Max epochs $U$, Loss weights $\alpha_{replay}$, $\alpha_{gip}$
**Output:** Parameter $\theta^{new}$ which can mitigate catastrophic forgetting of preceding classes
1: Initialize $\theta^{new}$ at random
2: **for** $u = 1$ to $U$ **do**
3:      $\mathcal{L}_{nc} = \ell_{ce}(Y^G, f(G; \theta^{new}))$
4:      $\mathcal{L}_{replay} = \ell_{ce}(Y^{\mathcal{M}}, f(\mathcal{M}; \theta^{new}))$
5:      $\mathcal{L}_{gip} = \ell_{reg}(f(\mathcal{M}; \theta^{old}), f(\mathcal{M}; \theta^{new}))$
6:      $\theta^{new} \leftarrow \arg\min \mathcal{L} = \mathcal{L}_{nc} + \alpha_{replay}\mathcal{L}_{replay} + \alpha_{gip}\mathcal{L}_{gip}$
7: **end for**
8: Add selected nodes to memory $\mathcal{M}$
9: **return** Parameter $\theta^{new}$

---

# B  Implementation Details

## B.1  Datasets

As illustrated in Table 5, we utilize five public datasets to evaluate the effectiveness of our proposed method in graph class incremental learning. Three different ways for partitioning datasets are employed: one involves an unequal division, where more classes are designated as base classes to enhance model robustness, while the remaining classes are treated as novel classes; the other two ways involve an equal division, with an equal number of classes allocated per task. The first dataset is CoraFull [48], which encompasses 70 classes. For the unequal division, 30 classes are used as base classes, and an additional 20 classes are selected as increments. Additionally, the classes are divided equally into either 10 or 2 classes per task. Arxiv [49] and Reddit [50], both of which consist of 40 classes. In the unequal division, 10 classes are designated as base classes, with increments of 5 classes. Similarly, the classes are evenly divided into either 10 or 2 classes per task. Each dataset has 3 tasks with 2 classes per task on Cora [51] and Citeseer [51].

Table 5: Statistics of datasets.

| Datasets | CoraFull | Arxiv | Reddit | Cora | Citeseer |
|---|---|---|---|---|---|
| # nodes | 19,793 | 169,343 | 227,853 | 2,708 | 3,327 |
| # edges | 130,622 | 1,166,243 | 114,615,892 | 5,429 | 4,732 |
| # class | 70 | 40 | 40 | 7 | 6 |
| # task | 3 / 7 / 35 | 7 / 4 / 20 | 7 / 4 / 20 | 3 | 3 |
| # base class | 30 / 10 / 2 | 10 / 10 / 2 | 10 / 10 / 2 | 2 | 2 |
| # novel class | 20 / 10 / 2 | 5 / 10 / 2 | 5 / 10 / 2 | 2 | 2 |

## B.2  Baselines

In this subsection, we introduce the baselines in the main paper. These baselines are as follows:

- **Finetuning** is the lower bound baseline updating the model only with newly emerging graph data.
- **Joint** is the ideal upper bound and inputs contain all previous graph data.

- Elastic Weight Consolidation (**EWC**) [30] quadratically penalizes model weights according to their importance to previous tasks.
- Gradient Episodic Memory (**GEM**) [31] modifies gradients of the current task using gradients computed from stored graph data.
- Memory Aware Synapses (**MAS**) [32] utilizes analysis of parameter prediction as the importance of parameters when adding regularization terms.
- Learning without Forgetting (**LwF**) [6] utilizes information distillation to reduce the discrepancy between old and new models.
- Topology-aware Weight Preserving (**TWP**) [1] preserves the key parameters and topology of the previous task through regularization terms.
- Structural Shift Risk Mitigation (**SSRM**) [11] introduces regularization terms to mitigate catastrophic forgetting from structural drift.
- Experience Replay Graph Neural Network (**ERGNN**) [7] framework incorporates memory replay by storing representative nodes.
- Sparsified Subgraph Memory (**SSM**) [8] stores sampled sparse subgraphs in memory repository to preserve structural information.
- The Condense and Train (**CaT**) [9] framework compresses the graph into a small but informative synthetic replay graph.

## B.3 Metrics

We choose two widely used metrics to evaluate the performance of the compared methods, including Average Performance (AP) and Average Forgetting (AF) [31]. When the model learns the latest task, all previous tasks are evaluated and a lower triangular performance matrix $W = \{w_{tt'}\} \in w^{\tau \times \tau}$ is formed, where $w_{tt'}$ is node classification accuracy on task $t$ after learning task $t'$ ($t \leq t'$) and $\tau$ is the total number of tasks. Average performance $\text{AP} = 1/\tau \sum_{t=1}^{\tau} w_{\tau,t}$ evaluates the average performance of model on previous task after learning from new task $\tau$. Average Forgetting $\text{AF} = 1/(\tau - 1) \sum_{t=1}^{\tau-1} (w_{\tau,t} - w_{t,t})$ represents the average performance degradation on previous tasks after learning from task $\tau$.

## B.4 Reproducibility

Our method is trained using a fixed random seed to ensure the consistency and verifiability of results. We are committed to open-sourcing and sharing our code to promote academic collaboration and knowledge sharing, enabling other researchers to reproduce and validate our experimental results.

Table 6: Incremental learning settings.

| | |
|---|---|
| ERGNN-GSIP | d: 0.5, sampler: CM |
| SSM-GSIP | subgraph_sampler: random |
| CaT-GSIP | n_encoders: 500, feat_init: randomChoice, feat_lr: 0.001, hid_dim: 512, hop: 1 |

## B.5 Detailed Settings

Our model is deployed in PyTorch with an NVIDIA RTX 3090 GPU and trained with 200 epochs for every task. We use Adam with weight decay for optimization, and the learning rate is set to 0.005. We use a two-layer GCN with a hidden dimension 256 as the backbone. All results are reported in means and standard deviations over 3 trials. The train-validation-test splitting ratios are 60%, 20%, and 20% for all datasets. The train-validation-test split is achieved through random sampling, resulting in variations in performance across different rounds of random sampling. $\widehat{\delta}$ is set to 0.99 and the search space of $\delta$ is $\{0.5, 0.6, 0.7, 0.8, 0.9\}$. Table 6 is the hyper-parameters we adopt from [42] and [9].

# C Experimental Results

## C.1 Performance Comparison

Table 7: Performance comparison on CoraFull, Arxiv, and Reddit for GCIL setting. Results are averaged among three trials. The best performing results (excluding Joint) are highlighted in **bold**, and the standard deviations are shown in gray.

| Method | CoraFull | | | | | | Arxiv | | | | | | Reddit | | | | | |
|---|---|---|---|---|---|---|---|---|---|---|---|---|---|---|---|---|---|---|
| | Unequally | | Equally (10) | | Equally (2) | | Unequally | | Equally (10) | | Equally (2) | | Unequally | | Equally (10) | | Equally (2) | |
| | AP↑ | AF↑ | AP↑ | AF↑ | AP↑ | AF↑ | AP↑ | AF↑ | AP↑ | AF↑ | AP↑ | AF↑ | AP↑ | AF↑ | AP↑ | AF↑ | AP↑ | AF↑ |
| Finetuning | 23.95 | -76.59 | 11.06 | -85.77 | 2.70 | -95.48 | 11.64 | -70.41 | 5.41 | -50.00 | 4.91 | -87.61 | 14.66 | -91.80 | 22.90 | -94.42 | 5.83 | -94.23 |
| | ±0.18 | ±0.80 | ±0.14 | ±0.03 | ±0.28 | ±0.04 | ±0.20 | ±0.76 | ±2.03 | ±3.93 | ±0.01 | ±0.41 | ±1.68 | ±3.51 | ±1.89 | ±1.74 | ±0.78 | ±1.32 |
| EWC | 24.09 | -75.78 | 11.15 | -86.08 | 5.13 | -93.08 | 11.93 | -68.97 | 14.83 | -57.33 | 4.91 | -87.58 | 13.79 | -95.35 | 22.30 | -95.27 | 9.66 | -93.85 |
| | ±0.48 | ±0.61 | ±0.25 | ±0.32 | ±1.99 | ±1.94 | ±0.15 | ±1.45 | ±0.31 | ±1.16 | ±0.01 | ±0.30 | ±0.27 | ±1.54 | ±0.49 | ±1.30 | ±2.03 | ±2.07 |
| GEM | 23.95 | -76.05 | 11.23 | -85.78 | 7.97 | -90.00 | 11.61 | -60.27 | 8.27 | -44.42 | 4.92 | -86.66 | 18.51 | -89.79 | 22.58 | -93.93 | 35.11 | -65.67 |
| | ±0.37 | ±0.63 | ±0.29 | ±0.07 | ±0.67 | ±0.67 | ±0.38 | ±5.38 | ±1.53 | ±1.49 | ±0.08 | ±0.61 | ±4.23 | ±6.06 | ±2.58 | ±1.08 | ±4.31 | ±4.57 |
| MAS | 24.20 | -75.97 | 10.94 | -82.37 | 4.43 | -89.22 | 11.09 | -66.76 | 12.32 | -57.99 | 5.29 | -81.64 | 15.45 | -0.50 | 25.54 | 0.01 | 5.98 | -14.17 |
| | ±0.59 | ±0.97 | ±0.48 | ±0.71 | ±0.64 | ±1.43 | ±0.23 | ±0.59 | ±0.69 | ±1.82 | ±0.83 | ±1.34 | ±3.38 | ±0.76 | ±1.41 | ±0.03 | ±1.82 | ±1.17 |
| LwF | 23.99 | -76.14 | 11.14 | -85.67 | 2.72 | -95.08 | 11.93 | -70.66 | 14.69 | -58.93 | 4.91 | -88.14 | 16.13 | -90.31 | 24.39 | -93.29 | 7.59 | -88.98 |
| | ±0.62 | ±0.55 | ±0.14 | ±0.63 | ±0.25 | ±0.05 | ±0.20 | ±0.74 | ±1.48 | ±1.25 | ±0.01 | ±0.09 | ±3.17 | ±3.99 | ±0.69 | ±0.60 | ±0.51 | ±0.74 |
| TWP | 23.86 | -75.74 | 11.01 | -85.43 | 3.56 | -94.66 | 11.93 | -69.26 | 14.41 | -56.56 | 4.90 | -87.75 | 13.95 | -96.17 | 21.22 | -96.41 | 9.34 | -94.24 |
| | ±0.17 | ±0.70 | ±0.21 | ±0.18 | ±1.02 | ±0.84 | ±0.08 | ±0.24 | ±0.54 | ±3.36 | ±0.01 | ±0.32 | ±0.90 | ±1.11 | ±1.58 | ±0.11 | ±3.46 | ±3.78 |
| SSRM | 63.62 | -16.24 | 31.39 | -60.61 | 3.22 | -89.29 | 31.51 | -45.12 | 26.61 | -46.22 | 26.16 | -61.24 | 78.40 | -20.92 | 76.78 | -23.16 | 83.96 | -15.41 |
| | ±0.56 | ±1.22 | ±2.25 | ±3.11 | ±0.29 | ±0.48 | ±0.44 | ±0.55 | ±0.11 | ±1.09 | ±0.85 | ±0.84 | ±7.42 | ±8.64 | ±4.35 | ±5.62 | ±3.04 | ±3.18 |
| ERGNN | 60.91 | -19.47 | 24.39 | -69.31 | 3.01 | -94.34 | 31.18 | -45.45 | 24.47 | -49.11 | 24.70 | -62.26 | 76.60 | -23.22 | 75.22 | -25.26 | 83.16 | -16.21 |
| | ±1.12 | ±1.43 | ±0.39 | ±0.73 | ±0.20 | ±0.51 | ±0.83 | ±1.75 | ±2.67 | ±3.03 | ±1.00 | ±0.75 | ±5.77 | ±6.47 | ±11.67 | ±14.13 | ±1.92 | ±2.06 |
| ERGNN-GSIP (Ours) | 67.22 | -10.91 | 71.15 | -11.37 | 44.79 | -44.60 | 34.09 | -32.59 | 33.88 | -27.97 | 40.21 | -28.96 | 90.82 | -6.05 | 89.59 | -2.03 | 93.03 | -5.50 |
| | ±0.44 | ±0.62 | ±0.98 | ±0.74 | ±1.77 | ±1.67 | ±0.77 | ±1.37 | ±0.87 | ±1.63 | ±0.92 | ±0.94 | ±0.70 | ±0.68 | ±2.04 | ±4.62 | ±3.87 | ±4.11 |
| SSM | 50.51 | -10.56 | 62.90 | -6.02 | 79.02 | -4.24 | 63.48 | -12.41 | 60.57 | -10.09 | 63.91 | -12.48 | 90.10 | -5.83 | 86.91 | -3.24 | 96.24 | -1.64 |
| | ±1.03 | ±0.94 | ±0.50 | ±0.37 | ±0.50 | ±0.23 | ±0.78 | ±0.01 | ±0.80 | ±1.19 | ±0.35 | ±0.58 | ±1.56 | ±1.14 | ±1.77 | ±0.56 | ±0.24 | ±0.31 |
| SSM-GSIP (Ours) | 55.32 | -2.50 | 63.86 | **0.08** | 79.31 | 0.70 | 63.36 | -7.27 | 61.34 | -6.34 | 64.16 | -8.87 | 90.74 | -3.97 | 87.41 | 0.13 | 96.25 | -0.65 |
| | ±0.75 | ±1.13 | ±0.85 | ±0.76 | ±0.50 | ±0.25 | ±1.13 | ±0.82 | ±0.77 | ±0.70 | ±0.37 | ±0.58 | ±0.44 | ±0.40 | ±1.60 | ±0.91 | ±0.37 | ±0.64 |
| CaT | 70.55 | -5.26 | 76.35 | -5.44 | 80.64 | -4.31 | **71.66** | -8.33 | 70.16 | -7.25 | 66.21 | -12.73 | **96.39** | -0.77 | 93.97 | -1.31 | **97.64** | -0.49 |
| | ±0.67 | ±0.40 | ±0.41 | ±0.58 | ±0.30 | ±0.43 | ±0.73 | ±0.26 | ±0.14 | ±0.79 | ±0.12 | ±0.09 | ±0.17 | ±0.40 | ±0.30 | ±0.08 | ±0.09 | ±0.04 |
| CaT-GSIP (Ours) | **71.06** | **-0.28** | **78.29** | -1.25 | **81.10** | **2.68** | 71.52 | **-4.76** | **70.57** | **-3.97** | **68.80** | **3.49** | 96.15 | **-0.23** | **94.23** | **0.21** | 97.55 | **1.04** |
| | ±0.54 | ±0.07 | ±0.11 | ±0.25 | ±0.18 | ±0.16 | ±0.64 | ±0.08 | ±0.14 | ±0.51 | ±0.24 | ±0.39 | ±0.30 | ±0.31 | ±0.28 | ±0.64 | ±0.05 | ±0.30 |
| Joint | 85.3 | - | 85.3 | - | 85.3 | - | 63.5 | - | 63.5 | - | 63.5 | - | 98.2 | - | 98.2 | - | 98.2 | - |
| | ±0.1 | | ±0.1 | | ±0.1 | | ±0.3 | | ±0.3 | | ±0.3 | | ±0.0 | | ±0.0 | | ±0.0 | |

The effect of GCIL on five datasets with standard deviation is presented in Table 7 and 8. GSIP improves existing information preservation methods under different dataset splitting scenarios. The performance matrices of SSM and CaT on CoraFull before and after incorporating GSIP are shown in Figure 8 and Figure 9. Remembering information from previous classes becomes challenging without the implementation of the proposed GSIP. The performance matrix illustrates a deceleration in the forgetting process after adopting the GSIP scheme. This is evident by the minimal changes and deepening in color of each column, indicating mitigation of the catastrophic forgetting problem. This is achieved through the preservation of information from previous model.

## C.2 Ablation Study

**Efficiency.** As shown in Table 9, Table 10, and Table 11, the consequences of ablation experiments with standard deviation for ERGNN, SSM, and CaT as baselines (B) are presented. We investigate the effectiveness of low-frequency local modules (LL), low-frequency global modules (LG), and high-frequency modules (H). The above components are added one by one to baselines for performance comparison. B is the baseline. B+LL indicates that it only uses low-frequency local preservation. B+LL+LG represents that it uses low-frequency local-global preservation. B+LL+LG+H is the full model, which uses low-frequency and high-frequency preservation. All the components are effective and can bring great benefits to the model in performance improvement.

Table 8: Performance comparison on Cora and Citeseer for GCIL setting. Results are averaged among three trials. The best performing results (excluding Joint) are highlighted in **bold**, and the standard deviations are shown in gray.

| Method | Cora | | Citeseer | |
|---|---|---|---|---|
| | Equally (2) | | Equally (2) | |
| | AP↑ | AF↑ | AP↑ | AF↑ |
| Finetuning | 32.58 | -96.83 | 31.46 | -77.86 |
| | ±0.00 | ±0.39 | ±0.27 | ±0.67 |
| EWC | 32.58 | -97.16 | 31.26 | -78.22 |
| | ±0.00 | ±0.55 | ±0.15 | ±0.61 |
| GEM | 32.70 | -97.12 | 31.39 | -77.70 |
| | ±0.19 | ±0.16 | ±0.09 | ±1.06 |
| MAS | 31.84 | -97.17 | 31.25 | -76.67 |
| | ±0.10 | ±0.22 | ±0.55 | ±1.18 |
| LwF | 32.58 | -97.57 | 31.44 | -78.29 |
| | ±0.00 | ±0.27 | ±0.00 | ±0.36 |
| TWP | 32.58 | -97.32 | 31.22 | -78.14 |
| | ±0.00 | ±0.17 | ±0.23 | ±0.22 |
| SSRM | 35.48 | -70.01 | 51.91 | -67.66 |
| | ±0.49 | ±1.12 | ±4.59 | ±6.67 |
| ERGNN | 65.48 | -46.09 | 47.65 | -51.12 |
| | ±0.76 | ±1.15 | ±0.57 | ±1.47 |
| ERGNN-GSIP (Ours) | 71.29 | -36.95 | 61.29 | -29.38 |
| | ±0.91 | ±1.22 | ±0.96 | ±2.28 |
| SSM | 67.64 | -19.78 | 60.99 | -13.60 |
| | ±3.14 | ±8.10 | ±1.43 | ±3.61 |
| SSM-GSIP (Ours) | 69.92 | -11.82 | 61.86 | **-8.39** |
| | ±1.46 | ±6.74 | ±2.23 | ±3.82 |
| CaT | 88.22 | -4.40 | 75.08 | -10.93 |
| | ±0.49 | ±1.04 | ±0.35 | ±1.07 |
| CaT-GSIP (Ours) | **89.60** | **1.84** | **77.02** | -9.95 |
| | ±0.62 | ±3.89 | ±0.70 | ±1.15 |
| Joint | 93.09 | - | 78.27 | - |
| | ±0.85 | | ±0.10 | |

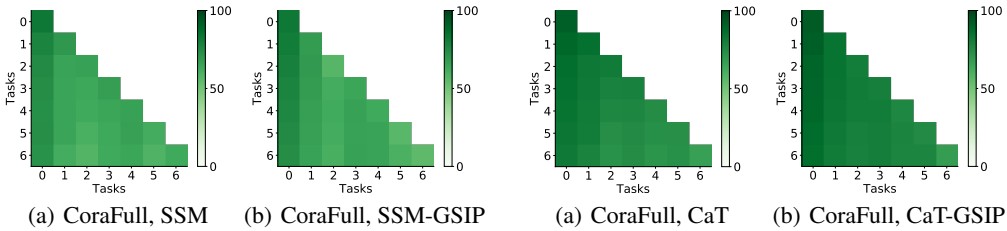

(a) CoraFull, SSM     (b) CoraFull, SSM-GSIP     (a) CoraFull, CaT     (b) CoraFull, CaT-GSIP

Figure 8: Performance matrices in SSM.     Figure 9: Performance matrices in CaT.

Table 9: Ablation comparisons of graph spatial information preserving strategy for ERGNN.

| Method | CoraFull | | | | | | Arxiv | | | | | | Reddit | | | | | |
|---|---|---|---|---|---|---|---|---|---|---|---|---|---|---|---|---|---|---|
| | Unequally | | Equally (10) | | Equally (2) | | Unequally | | Equally (10) | | Equally (2) | | Unequally | | Equally (10) | | Equally (2) | |
| | AP↑ | AF↑ | AP↑ | AF↑ | AP↑ | AF↑ | AP↑ | AF↑ | AP↑ | AF↑ | AP↑ | AF↑ | AP↑ | AF↑ | AP↑ | AF↑ | AP↑ | AF↑ |
| B | 60.91 | -19.47 | 24.39 | -69.31 | 3.01 | -94.34 | 31.18 | -45.45 | 24.47 | -49.11 | 24.70 | -62.26 | 76.60 | -23.22 | 75.22 | -25.26 | 83.16 | -16.21 |
| | ±1.12 | ±1.43 | ±0.39 | ±0.73 | ±0.20 | ±0.51 | ±0.83 | ±1.75 | ±2.67 | ±3.03 | ±1.00 | ±0.75 | ±5.77 | ±6.47 | ±11.67 | ±14.13 | ±1.92 | ±2.06 |
| B+LL | 65.79 | -13.20 | 69.02 | -14.15 | 41.37 | -47.39 | 33.27 | -34.60 | 27.10 | -40.95 | 38.09 | -35.04 | 84.63 | -12.09 | 84.26 | -12.37 | 87.52 | -10.97 |
| | ±0.87 | ±1.26 | ±0.11 | ±0.05 | ±1.90 | ±2.19 | ±1.48 | ±1.29 | ±3.47 | ±4.17 | ±1.02 | ±1.67 | ±4.58 | ±6.36 | ±1.12 | ±1.96 | ±2.86 | ±2.98 |
| B+LL+LG | 66.22 | -12.78 | 69.77 | -13.13 | 41.84 | -46.73 | 34.00 | -33.61 | 32.80 | -34.40 | 39.89 | -28.85 | 89.21 | -6.97 | 87.17 | -9.45 | 91.34 | -7.12 |
| | ±0.56 | ±1.12 | ±0.37 | ±0.29 | ±1.25 | ±1.16 | ±0.37 | ±0.58 | ±2.71 | ±3.17 | ±0.16 | ±0.49 | ±2.43 | ±0.94 | ±2.70 | ±3.86 | ±3.58 | ±3.91 |
| B+LL+LG+H | **67.22** | **-10.91** | **71.15** | **-11.37** | **44.79** | **-44.60** | **34.09** | **-32.59** | **33.88** | **-27.97** | **40.21** | -28.96 | **90.82** | **-6.05** | **89.59** | **-2.03** | **93.03** | **-5.50** |
| | ±0.44 | ±0.62 | ±0.98 | ±0.74 | ±1.77 | ±1.67 | ±0.77 | ±1.37 | ±0.87 | ±1.63 | ±0.92 | ±0.94 | ±0.70 | ±0.68 | ±2.04 | ±4.62 | ±3.87 | ±4.11 |

Table 10: Ablation comparisons of graph spatial information preserving strategy for SSM.

| Method | CoraFull | | | | | | Arxiv | | | | | | Reddit | | | | | |
|---|---|---|---|---|---|---|---|---|---|---|---|---|---|---|---|---|---|---|
| | Unequally | | Equally (10) | | Equally (2) | | Unequally | | Equally (10) | | Equally (2) | | Unequally | | Equally (10) | | Equally (2) | |
| | AP↑ | AF↑ | AP↑ | AF↑ | AP↑ | AF↑ | AP↑ | AF↑ | AP↑ | AF↑ | AP↑ | AF↑ | AP↑ | AF↑ | AP↑ | AF↑ | AP↑ | AF↑ |
| B | 50.51 | -10.56 | 62.90 | -6.02 | 79.02 | -4.24 | **63.48** | -12.41 | 60.57 | -10.09 | 63.91 | -12.48 | 90.10 | -5.83 | 86.91 | -3.24 | 96.24 | -1.64 |
| | ±1.03 | ±0.94 | ±0.50 | ±0.37 | ±0.50 | ±0.23 | ±0.78 | ±0.01 | ±0.80 | ±1.19 | ±0.80 | ±0.58 | ±1.56 | ±1.14 | ±1.77 | ±0.56 | ±0.24 | ±0.31 |
| B+LL | 54.75 | -5.45 | 63.68 | -1.60 | 79.30 | -0.49 | 62.99 | -9.70 | 60.13 | -7.04 | 63.93 | -9.89 | 90.55 | -4.06 | 87.34 | 0.06 | 95.98 | -0.84 |
| | ±0.76 | ±1.06 | ±0.58 | ±0.60 | ±0.65 | ±0.36 | ±0.91 | ±0.18 | ±0.48 | ±0.98 | ±0.34 | ±0.58 | ±0.48 | ±0.46 | ±1.56 | ±0.93 | ±0.19 | ±0.30 |
| B+LL+LG | 55.16 | -2.62 | 63.67 | -0.02 | **79.31** | 0.61 | 63.40 | -7.60 | 61.17 | -6.57 | 64.01 | -9.22 | 90.70 | -4.05 | 87.38 | **0.14** | 96.16 | -0.75 |
| | ±0.56 | ±1.21 | ±0.65 | ±0.59 | ±0.65 | ±0.60 | ±1.16 | ±0.96 | ±1.02 | ±1.32 | ±0.60 | ±1.07 | ±0.35 | ±0.33 | ±1.69 | ±0.93 | ±0.35 | ±0.57 |
| B+LL+LG+H | **55.32** | **-2.50** | **63.86** | **0.08** | **79.31** | **0.70** | 63.36 | **-7.27** | **61.34** | **-6.34** | **64.16** | **-8.87** | **90.74** | **-3.97** | **87.41** | 0.13 | **96.25** | **-0.65** |
| | ±0.75 | ±1.13 | ±0.85 | ±0.76 | ±0.50 | ±0.25 | ±1.13 | ±0.82 | ±0.77 | ±0.70 | ±0.37 | ±0.58 | ±0.44 | ±0.40 | ±1.60 | ±0.91 | ±0.37 | ±0.64 |

Table 11: Ablation comparisons of graph spatial information preserving strategy for CaT.

| Method | CoraFull | | | | | | Arxiv | | | | | | Reddit | | | | | |
|---|---|---|---|---|---|---|---|---|---|---|---|---|---|---|---|---|---|---|
| | Unequally | | Equally (10) | | Equally (2) | | Unequally | | Equally (10) | | Equally (2) | | Unequally | | Equally (10) | | Equally (2) | |
| | AP↑ | AF↑ | AP↑ | AF↑ | AP↑ | AF↑ | AP↑ | AF↑ | AP↑ | AF↑ | AP↑ | AF↑ | AP↑ | AF↑ | AP↑ | AF↑ | AP↑ | AF↑ |
| B | 70.55 | -5.26 | 76.35 | -5.44 | 80.64 | -4.31 | **71.66** | -8.33 | 70.16 | -7.25 | 66.21 | -12.73 | **96.39** | -0.77 | 93.97 | -1.31 | **97.64** | -0.49 |
| | ±0.67 | ±0.40 | ±0.41 | ±0.58 | ±0.30 | ±0.43 | ±0.73 | ±0.26 | ±0.14 | ±0.79 | ±0.12 | ±0.09 | ±0.17 | ±0.40 | ±0.30 | ±0.08 | ±0.09 | ±0.04 |
| B+LL | 70.74 | -1.32 | **78.31** | -2.03 | 80.91 | -0.89 | 71.42 | -6.02 | 70.42 | -4.80 | 67.23 | -2.85 | 96.12 | -0.29 | 93.23 | -0.12 | 97.26 | 0.35 |
| | ±0.57 | ±0.69 | ±0.16 | ±0.13 | ±0.14 | ±0.52 | ±0.77 | ±0.20 | ±0.37 | ±0.75 | ±0.24 | ±0.44 | ±0.28 | ±0.31 | ±0.39 | ±0.78 | ±0.28 | ±0.36 |
| B+LL+LG | 70.89 | -0.39 | 77.91 | -1.41 | **81.12** | -0.10 | 71.41 | -4.85 | **70.59** | -3.99 | 68.61 | 3.10 | 96.14 | -0.26 | 93.91 | 0.04 | 97.27 | 0.48 |
| | ±0.42 | ±0.07 | ±0.27 | ±0.33 | ±0.19 | ±0.17 | ±0.92 | ±0.04 | ±0.15 | ±0.59 | ±0.10 | ±0.34 | ±0.24 | ±0.24 | ±0.47 | ±1.09 | ±0.33 | ±0.36 |
| B+LL+LG+H | **71.06** | **-0.28** | 78.29 | **-1.25** | 81.10 | 2.68 | 71.52 | **-4.76** | 70.57 | **-3.97** | **68.80** | 3.49 | 96.15 | **-0.23** | **94.23** | **0.21** | 97.55 | **1.04** |
| | ±0.54 | ±0.07 | ±0.11 | ±0.25 | ±0.18 | ±0.16 | ±0.64 | ±0.08 | ±0.14 | ±0.51 | ±0.24 | ±0.39 | ±0.30 | ±0.31 | ±0.28 | ±0.64 | ±0.05 | ±0.30 |

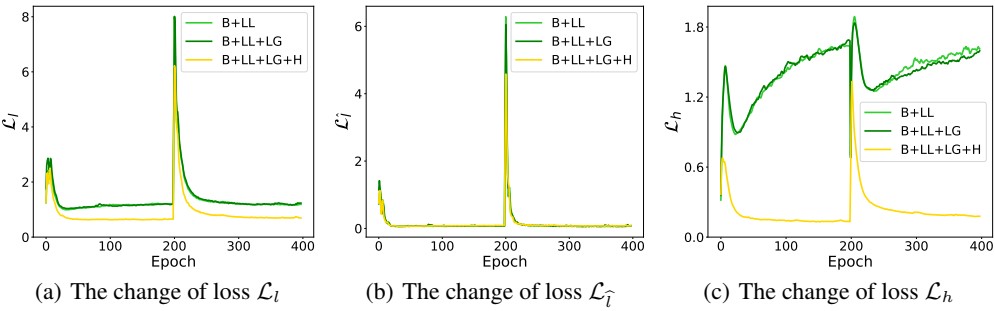

(a) The change of loss $\mathcal{L}_l$     (b) The change of loss $\mathcal{L}_{\hat{l}}$     (c) The change of loss $\mathcal{L}_h$

Figure 10: Graph information preservation training losses of different variants with epochs on CoraFull dataset.

**Information Pattern.** Training losses of ERGNN-GSIP during incremental processes in the dataset inequality partition setting are illustrated. Specifically, we focus on the last two tasks and examine how the loss of different variants changes with increasing epochs. We make three observations: (1) Low-frequency local graph information is well preserved. Figure 10(a) measures the degree of feature similarity. It can be observed that losses of the first two variants start to rise around the 50th epoch. However, the GSIP loss remains much lower than the first two variants and converges quickly. (2) The learning of low-frequency information is ensured after global graph embedding similarity correction. It can be seen that the loss of GSIP is slightly lower than the other two variants in Figure 10(b). (3) GSIP can preserve high-frequency information and reduce the forgetting of topology. In Figure 10(c), we can see that the losses of the first two variants decrease and then increase at each increment. This demonstrates that node difference information is almost completely discarded in these variants.

## C.3 Hyper-Parameter Analysis of Memory Size

We analyze the impact of the number of storage nodes for each task $\#\mathcal{M}$ on performance of each task that has 2 classes. As depicted in Figure 11-Figure 15, it can be observed that the proposed method consistently outperforms the original method in terms of the AP (the higher, the better) and -AF metric (the lower, the better), regardless of the value of $\#\mathcal{M}$. CaT cannot be trained with 400 nodes on CoraFull due to Cuda memory constraints. Interestingly, despite having less memory, the proposed method demonstrates superior performance across three datasets. Furthermore, the performance remains relatively consistent when storing 200 or 400 nodes, indicating that our method does not incur higher storage costs. Notably, storing only 10 nodes per task yields performance comparable to storing 400 nodes on Reddit, highlighting the superiority of our approach.

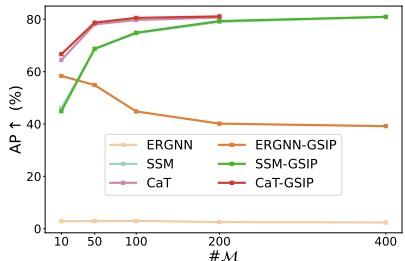

Figure 11: The change of AP affected by $\#\mathcal{M}$ on CoraFull dataset.

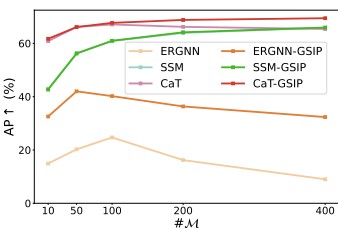

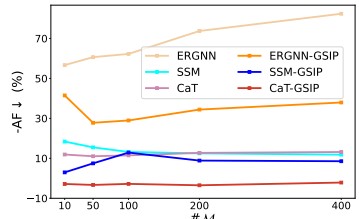

Figure 12: The change of AP affected by $\#\mathcal{M}$ on Arxiv dataset.

Figure 13: The change of AF affected by $\#\mathcal{M}$ on Arxiv dataset.

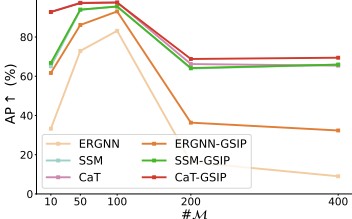

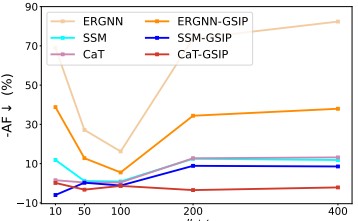

Figure 14: The change of AP affected by $\#\mathcal{M}$ on Reddit dataset.

Figure 15: The change of AF affected by $\#\mathcal{M}$ on Reddit dataset.

## C.4 Hyper-Parameter Analysis of Loss Weights

As shown in Figure 16, Figure 17, and Figure 18, the analysis for two hyper-parameters, loss weights $\beta$ and $\gamma$ on ERGNN, SSM, and CaT in terms of AF is conducted, and the results are in an experimental setting of increment 2. For ERGNN, $\beta_1$ is set to $[2e-5, 1e-1, 1e-5]$ and $\gamma_1$ is set to $[2e-2, 1e-8, 1e-4]$ for different datasets. For SSM, $\beta_1$ is set to $[1, 1, 10]$ and $\gamma_1$ is set to $[1e-7, 1e-8, 1e-3]$ for three datasets. For CaT, $\beta_1$ is set to $[1e-1, 2, 2e-4]$ and $\gamma_1$ is set to $[1e-1, 1e-6, 2e-1]$ for three datasets. We notice that the model performance remains unaffected by changes in $\beta$ when $\gamma$ is set to 0 and by changes in $\gamma$ when $\beta$ reaches its optimal value.

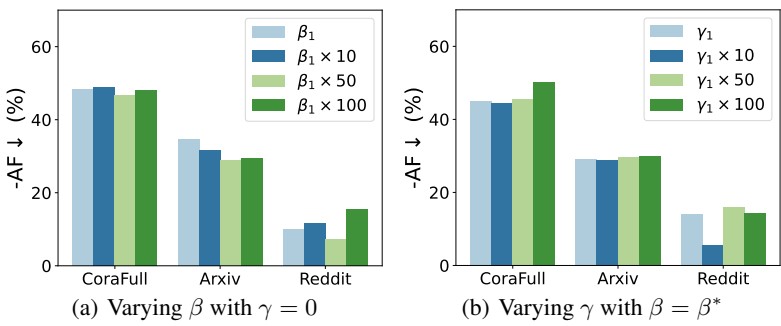

Figure 16: The analysis of $\beta$ and $\gamma$ in ERGNN-GSIP on CoraFull, Arxiv, and Reddit datasets.

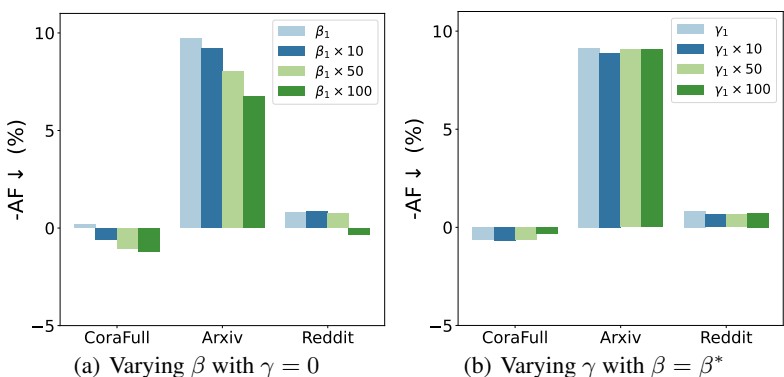

Figure 17: The analysis of $\beta$ and $\gamma$ in SSM-GSIP on CoraFull, Arxiv, and Reddit datasets.

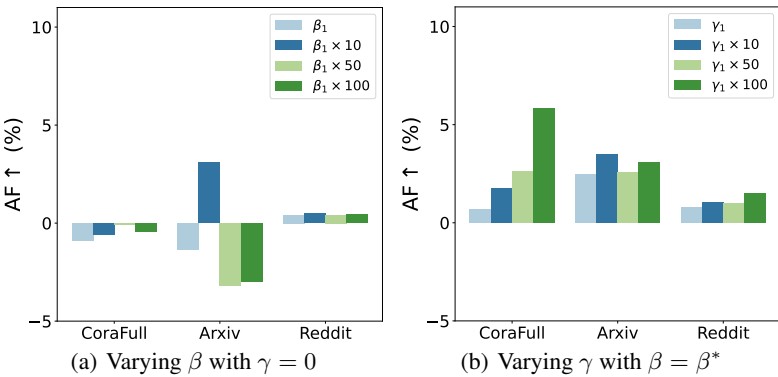

Figure 18: The analysis of $\beta$ and $\gamma$ in CaT-GSIP on CoraFull, Arxiv, and Reddit datasets.

### C.5 Visualization

To qualitatively demonstrate the effectiveness of representations, we utilize t-SNE [52] to visualize the node embeddings of ERGNN and ERGNN-GSIP. After learning the last task, Figure 19(a) and Figure 19(b) display the results of the learned node embeddings in Task 1 on Reddit, while Figure 19(c) and Figure 19(d) illustrate the results of the last task. GSIP exhibits superior representation ability, effectively considering representations and accurately classifying old and new classes.

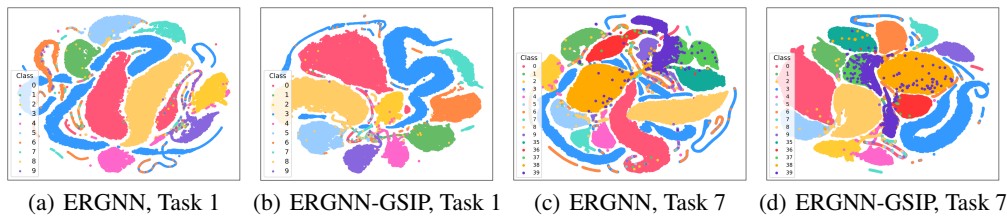

|(a) ERGNN, Task 1 | (b) ERGNN-GSIP, Task 1 | (c) ERGNN, Task 7 | (d) ERGNN-GSIP, Task 7|

Figure 19: The visualization of node embeddings from ERGNN and ERGNN-GSIP in Task 1 and Task 7 of Reddit dataset.

### C.6 Time Complexity Analysis

The time complexity of framework mainly comes from three aspects: (1) node classification $O(|V|\Bbbk^G\Bbbk + |\mathcal{E}|\Bbbk + |V|\Bbbk)$, (2) replay scheme $O(|\mathcal{M}|\Bbbk^G\Bbbk + |\mathcal{E}^{\mathcal{M}}|\Bbbk + |\mathcal{M}|\Bbbk)$, and (3) graph spatial information preservation $O(|\mathcal{E}^{\mathcal{M}}|\Bbbk + \Bbbk)$. The total time complexity is $O(|V|\Bbbk^G\Bbbk + |\mathcal{M}|\Bbbk^G\Bbbk + |V|\Bbbk + |\mathcal{M}|\Bbbk + |\mathcal{E}|\Bbbk + |\mathcal{E}^{\mathcal{M}}|\Bbbk + \Bbbk)$, where $V$ and $\mathcal{M}$ are subgraph vertices and memory, $\mathcal{E}$ and $\mathcal{E}^{\mathcal{M}}$ are edge sets of subgraph vertices and memory, then $\Bbbk^G$ and $\Bbbk$ denote the dimensions of inputs and hidden spaces. The time complexity increases linearly compared with baselines. The running time on CoraFull is presented in Table 12. For ERGNN, $|V| \gg |\mathcal{M}|$ leads to phenomenon that the higher the number of tasks, the shorter the training time. On the contrary, more tasks result in longer training time due to the properties of compressed graphs in SSM and CaT.

Table 12: Running time (s) of each epoch under three dataset partitioning cases on CoraFull dataset.

| Method | Unequally | Equally (10) | Equally (2) |
|--------|-----------|--------------|-------------|
| ERGNN | 0.8073 | 0.3694 | 0.1660 |
| **+GSIP** | 1.1913 | 0.4672 | 0.3792 |
| SSM | 0.0110 | 0.0279 | 0.0410 |
| **+GSIP** | 0.0113 | 0.0312 | 0.0494 |
| CaT | 0.0106 | 0.0189 | 0.0488 |
| **+GSIP** | 0.0147 | 0.0235 | 0.0524 |

## D Discussion

### D.1 Limitation

The primary limitation of GSIP lies in its focus on replayed designs and the lack of connection to other methods. Also, the paper does not examine other GCIL settings on the graph, such as the lack of a clear task boundary, which would be an interesting direction to explore in the future.

### D.2 Broader Impact

Considering the broader implications of our work, we posit that the proposed framework for information preservation of the old graph model will support the development of systems in an open environment based on machine learning. However, accessing old data may raise privacy concerns, and the dynamic updating of systems could inadvertently marginalize under-represented groups, potentially have a negative impact on outcomes and interfere with fair decision-making.

