# OpenReview forum: "What Matters in Graph Class Incremental Learning? An Information Preservation Perspective"
_NeurIPS.cc/2024/Conference — NeurIPS 2024 poster_

### Official Review · Reviewer_T3gk · 2024-07-10

**Soundness:** 3
**Presentation:** 2
**Contribution:** 2
**Rating:** 6
**Confidence:** 3

**Summary:**

This paper studies graph class incremental learning (GCIL), which requires the model to classify emerging nodes of new classes while remembering old classes. The paper provides a theoretical analysis of GCIL and finds that preserving old graph information that corresponds to local-global low-frequency and high-frequency components in the spatial domain can calibrate semantic and structural shifts to reduce catastrophic forgetting risk. Then, the paper proposes a method to utilise node representations on old and new models to preserve node features, graphs, and neighbor distances.

**Strengths:**

1. The paper studies graph class incremental learning (GCIL), addressing the problem of supervised node classification within the context of an expanding graph. The motivation is interesting, and the challenge really exists in the real-world graph tasks.
2. The paper gives a theoretical analysis of GCIL and divides the framework into low and high-frequency modules to preserve old information.
3. The paper evaluates the proposed method on three public datasets and the better results compared with baselines show the effectiveness of the proposed method.

**Weaknesses:**

1. The motivation for preserving global low-frequency information about the graph is uncleared. Why only separate local and global components for low-frequency and not separate local and global components for high-frequency? More theoretical analysis and studies are needed.
2. Why is the experimental setting for each dataset's tasks configured as described in the paper? Is it following previous research works? It is highly recommended to provide results for different task settings.
3. It is highly recommended to add the implementation details, especially the hyper-parameter for each dataset in the paper.
4. More hyper-parameter experiments about the method are needed. For example, the parameter sensitivity analysis about the \alpha.

**Questions:**

Please see the above weakness.

**Limitations:**

The paper presents several limitations: The primary limitation of GSIP lies in its focus on replayed designs and the lack of connection to other methods. Also, the paper does not examine other GCIL settings on the graph, such as the lack of a clear task boundary, which would be an interesting direction to explore in the future.

---

> ### Author Rebuttal · Authors · 2024-08-06
>
> Thanks for your positive comments!
>
> >**Q1: Why only separate local and global components for low-frequency and not separate local and global components for high-frequency?**
>
> A1: The rationale for maintaining low-frequency global information (LG) without high-frequency global information (HG) is that the design of HG is redundant, incurs high computational costs, and does not yield performance improvements.
>
> 1. **Redundant method design and higher time complexity**.
> * We need to clarify that LG is meaningful as it represents overall graph information contained in each category dimension. It can be instantiated using graph pooling and does not need to waste much time.
>
> * The general formula for HG preservation is
> $$\left\|\Delta{\mathbf{z}^{\widehat{h}}_ i}\right\|_ 2^2 = \left\|\left({z}_ i^{old} - \sum_ {j \in \mathcal{M}} \frac{{z}_ j^{old}}{\sqrt{\left|\mathcal{M}\right|^2}}\right) - \left({z}_ i^{new} - \sum_ {j \in \mathcal{M}} \frac{{z}_ j^{new}}{\sqrt{\left|\mathcal{M}\right|^2}}\right)\right\|_ 2^2.$$ HG implies that every node in replayed graph needs to calculate disparity with other nodes. Since GNN assumes that neighboring nodes have similar representations, penalizing distance between nodes and their neighbors at multiple hops away is redundant and does not contribute to structural preservation. Moreover, the inclusion of this term imposes an optimization burden and exhibits high time complexity ($O(|\mathcal{M}|^2 \cdot {k})$, ${k}$ is dimensions of hidden spaces).
>
>
> 2. **Lower experimental result.**
> * According to ablation experiment in Table 2, the preservation of LG in ERGNN-GSIP on CoraFull result in an increase of 0.75\% in AP and 1.02\% in AF. On Reddit, the preservation of LG in ERGNN-GSIP led to an increase in AP by 4.58\%, 2.91\%, and 3.82\%, and in AF by 5.12\%, 2.92\%, and 3.85\% across three task divisions.
>
> * We conduct experiments on CoraFull and find that the addition of HG preservation loss does not lead to performance improvement.
>
> | |Unequally| |Equally (10)| |Equally (2)| |
> |:----|:----|:----|:----|:----|:----|:----|
> | |AP|AF|AP|AF|AP|AF|
> |GSIP|**67.22±0.44** |-10.91±0.62|**71.15±0.98**|**-11.37±0.74**|**44.79±1.77**|**-44.60±1.67**|
> |+HG|66.63±0.49|**-10.07±1.75**|69.96±0.85|-12.85±0.94|24.66±±1.47|-68.48±1.35|
>
>
> >**Q2: Why is the experimental setting for each dataset's tasks configured as described in the paper?**
>
> A2: We adopt standard configuration from prior studies [1][2][3][4], which is the current mainstream setup, and integrate two new task setups. Additional configurations have contributed to a reduction in variance. Subsequently, we conduct tests on two new datasets within the general setup.
>
> 1. **Task settings as previous works + New task configuration.** We clarify that existing works are set with 2 classes per task. We have followed the setup of previous works [1][2][3][4], corresponding to our setup of Equally (2). We introduce two new settings: Equally (10) and Unequally.
>
> 2. **New task configuration helps to reduce variance.** We believe that  new task configurations can bring a certain level of robustness. As shown in Table 5 on Page 16, On Reddit, the variances of AP and AF for Equally (2) reach 3.87 and 4.11 for ERGNN-GSIP, while for Unequally, the variances of AP and AF are only 0.70 and 0.68, with a reduction of more than 3.
>
> 3. **New experimental results.** As shown in Table R1 of PDF file, we have added results on the Cora and Citeseer datasets with an increment of 2 and a task number of 3, validating the excellence of our method.
>
> [1] Xikun Zhang et al. CGLB: Benchmark tasks for continual graph learning. NeurIPS 2022.
>
> [2] Yilun Liu et al. Cat: Balanced continual graph learning with graph condensation. ICDM 2023.
>
> [3] Junwei Su et al. Towards robust graph incremental learning on evolving graphs. ICML 2023.
>
> [4] Zeyang Zhang et al. Disentangled Continual Graph Neural Architecture Search with Invariant Modular Supernet. ICML 2024.
>
>
> >**Q3: It is highly recommended to add the implementation details, especially the hyper-parameter for each dataset in the paper.**
>
> A3: **We summarize experimental setup and hyper-parameters used for each dataset.**
>
> 1. **Experimental setup.** Each task undergoes a training regimen of 200 epochs. For optimization, we employ the Adam algorithm with weight decay, setting the learning rate to a value of 0.005. The model's architecture is anchored by a two-layer Graph Convolutional Network (GCN) featuring a hidden dimension of 256. The datasets are partitioned into train-validation-test splits with ratios of 60\%, 20\%, and 20\%, respectively.
>
> 2. **Hyper-parameter.** The settings of three scaling factors $\alpha_{gip}$ (the weights of $\mathcal{L}_{gip}$), $\beta$, and $\gamma$ on each dataset are shown in Table R2 of PDF file, where U, E(10), and E(2) are three task partitioning modes, the first/middle/last three rows are ERGNN, SSM, and CaT.
>
>
> >**Q4: More hyper-parameter experiments about the method are needed. For example, the parameter sensitivity analysis about the $\alpha$.**
>
> A4:  As shown in Figure R3 of PDF file, we conduct the analysis for $\alpha$ (has changed to $\alpha_{replay}$) with ERGNN, SSM, and CaT on three datasets.
>
> $\alpha_{replay,1}$ is set to $\alpha_{replay,o} \times 0.1$ for three methods, the hyper-parameters we used are all $\alpha_{replay,1} \times 10$ (i.e., $\alpha_{replay,o}$, dark blue bars in Figure R3). $\alpha_{replay}$ is the hyper-parameter designed in the baseline and $\alpha_{replay,o}$ is the original setting.
>
> Although this setting may not be optimal for performance, we still followed the baseline's setting for a **fair comparison**. In ERGNN, the greater the loss weight, the better the performance, possibly because ERGNN focuses too much on these independent replay nodes. In SSM and CaT, except when $\alpha_{replay,1}$ is used, performance hardly changes with an increase in the loss weight, indicating that $\alpha_{replay}$ is not sensitive.

---

### Official Review · Reviewer_BwJ9 · 2024-07-11

**Soundness:** 3
**Presentation:** 1
**Contribution:** 3
**Rating:** 5
**Confidence:** 4

**Summary:**

This paper studies the graph class incremental learning problem, and specially focuses on theoretically investigating what matters in preserving the information from the old classes.

The authors theoretically demonstrate that maintaining the graph information can preserve information of the old model, such that the node semantic and graph structure shift can be overcome. Based on this finding, the authors split the graph information into low- and high frequency parts, and designed alignment based techniques to preserve these two types of information.

The proposed GSIP can be integrated with different baseline methods, and the empirical results on three datasets demonstrated the effectiveness of GSIP.

**Strengths:**

1. The proposed work includes both theoretical foundations and empirical improvements.

2. The semantic shift and structural shift are quantified and demonstrated on CoraFull dataset, which makes the mechanism of the forgetting more tangible.

**Weaknesses:**

1. The writing requires improvements. The abstract and introduction do not provide a clear overview of the work, since there are multiple unclear terms like graph information and local-global parts. I would recommend the authors to revise this part. At least these terms should be briefly introduced in Introduction before being used.

2. The notations are not consistent. In 4.1, the bold Z-old and Z-new seem to be same as the Z-old and Z-new above in Section 3, but are in different fonts.

**Questions:**

1. What is the rationale behind the definition of the structural shift score? Does this implies the difference between each node and its neighbors?

2. The forgetting is demonstrated through the node semantic shift and the structural shift, why is the following analysis conducted from the low- and high-frequency perspectives?

**Limitations:**

The authors have discussed limitations regarding the method design, positioning of the work against other baselines, and empirical settings.

---

> ### Author Rebuttal · Authors · 2024-08-06
>
> Thanks for your positive comments!
>
> >**Q1: The writing requires improvements. The abstract and introduction do not provide a clear overview of the work, since there are multiple unclear terms like graph information and local-global parts. I would recommend the authors to revise this part. At least these terms should be briefly introduced in the Introduction before being used.**
>
> A1: Thanks for your comments. We will provide a brief introduction to some terms before using them, such as defining "graph information" as "Information in the graph data containing node semantic information and graph topological information", and changing "local-global parts" to "local-global information".
>
>
> >**Q2: The notations are not consistent. In 4.1, the bold Z-old and Z-new seem to be the same as the Z-old and Z-new above in Section 3 but are in different fonts.**
>
> A2: Thank you for the comments. We will modify the font format of bold Z-old and Z-new in Section 4.1 for easier distinction. They have different meanings, the bold Z-old and Z-new in Section 4.1 represent old graph information and new graph information, and Z-old and Z-new in Section 3 represent the node representations of the old model and the new models.
>
>
> >**Q3: What is the rationale behind the definition of the structural shift score? Does this imply the difference between each node and its neighbors?**
>
> A3: We utilize the structural shift score to quantitatively measure the degree of forgetting of the topological structure learned by the new model compared to that learned by the old model. We have to emphasize that the structural shift score implies the gap between the differences between nodes and their neighbors in the new and old models, rather than merely measuring the differences between nodes and their neighbors.
>
> **The rationale behind the definition of structural shift score comes from two aspects:**
>
> 1. **Structural shift score is an effective way to quantitatively measure catastrophic forgetting from topological structure when learning new tasks.** If the new model can effectively mimic the topological structure of the old model, then forgetting can be mitigated. Firstly, we utilize node representations from old and new models to infer topological structure, and similar features suggest potential edges connecting nodes. Subsequently, we employ the Anonymous Walk (AWE) [1], which reflects topological information, to obtain representations of inferred old graph and new graph. Finally, we measure the difference in graph structure representations using cosine similarity as structural shift score, and the greater the difference, the higher the score. The real topological structure is not targeted due to potential noise [2].
>
> 2. **The structure shift score is used as a metric to evaluate structural shift.** As shown in Figure 2 on Page 2, structural shift does indeed exist during the incremental process, and the structural shift becomes increasingly larger as new tasks are learned. As shown in Figure 5(c) on Page 8, the structural shift is reduced to almost near 0, indicating that our method has well-calibrated the structural shift.
>
> [1] Sergey Ivanov et al. Anonymous Walk Embeddings. ICML 2018.
>
> [2] Seungyoon Choi et al. DSLR: diversity enhancement and structure learning for rehearsal-based graph continual learning. WWW 2024.
>
> >**Q4: The forgetting is demonstrated through the node semantic shift and the structural shift, why is the following analysis conducted from the low- and high-frequency perspectives?**
>
> A4: Graph information encompasses both feature information and topological information. We utilize the analysis of graph information preservation and the decomposition of graph information to map the preservation of graph information to the spatial domain and instantiate it. Low-frequency information preservation mitigates feature shift, while high-frequency information preservation alleviates topological shift, which has been validated through experimental verification.
>
> 1. **Low-frequency information corresponds to feature information.** Low-frequency information represents the sum of nodes and their neighbors, which is similar to the process of message passing and information aggregation in graph neural networks to obtain higher-order node features. Preserving low-frequency information is equivalent to aligning higher-order features formed by general GNN and thus effectively mitigating node semantic shift.
>
> 2. **High-frequency information corresponds to topological information.** High-frequency information represents the differences between nodes and their neighbors, maintaining the graph structure information through the support of the old model. The distance between a node and its neighbor is a measure of whether an edge exists. The closer the distance between the node and its neighbors, it means that there is a high probability that there is an edge, and vice versa.
> The old model implies the graph information of the old data because it has seen the old graph data during training. We utilize the similarity between nodes and their neighboring nodes from the old model's output and then optimize the new model to imitate the structural similarity distribution of the old model. High-frequency information preservation fortunately addresses the challenge of topological shift.
>
> 3. **Experimental verification.** As can be seen from Figure 5(c) on Page 8, the semantic shift score and the structure shift score are almost close to 0, thus semantic and topological shifts are well calibrated through the preservation of low-frequency and high-frequency information.

---

### Official Review · Reviewer_9jHn · 2024-07-12

**Soundness:** 3
**Presentation:** 2
**Contribution:** 3
**Rating:** 6
**Confidence:** 4

**Summary:**

This paper proposes an innovative framework named graph spatial information preservation (GSIP), which alleviates catastrophic forgetting in graph class incremental learning (GCIL), by preserving low-frequency local-global information and high-frequency information in both the feature space and the topological space.

**Strengths:**

Strength 1: This paper has good originality to identify a unique challenge in GCIL task, which is the lack of theoretical understanding of information preservation.



Strength 2: This paper provides detailed mathematical derivations to explain how catastrophic forgetting can be alleviated by preserving graph information in the spatial domain.



Strength 3: This paper conducts comprehensive experiments, along with ablation studies, hyper-parameter tuning analyses, and case studies, to demonstrate the effectiveness of the proposed framework.

**Weaknesses:**

Weakness 1: The generalizability regarding Figure 1 could be further improved.

Weakness 2: The writing in terms of clarity and conciseness could be further improved.

Weakness 3: It is a bit confusing to take another average (MSE) immediately after an average (mean pooling). The reasoning for Equation 14 could be better clarified with more details.

**Questions:**

Question 1: From line 50 to line 52, how to quantitatively understand “a larger distortion”? According to Figure 1, does “a larger distortion” indicate that the black dotted ellipse in Figure 1 (a) has a longer major axis than that in Figure 1 (c)? If so, since the five nodes are randomly selected and connected, then what about other red nodes having the same class but not included within the black dotted ellipse?




Question 2: Do the parameter isolation methods, the replay methods, and the regulation methods mentioned in the Introduction section (from line 28 to line 37), have the same meaning as those mentioned in the Related Work section (from line 83 to line 89)?




Question 3: From line 216 to line 217, based on Equation 14, since the mean pooling has been applied, what is the motivation and reasoning to introduce the mean squared error (MSE) loss to evaluate global representation gaps?

**Limitations:**

The authors have not adequately addressed the limitations and the potential negative societal impact of their work.

Suggestion for improvement: Data privacy should be discussed.

---

> ### Author Rebuttal · Authors · 2024-08-06
>
> Thanks for your valuable suggestions!
>
> >**Q1: From line 50 to line 52, how to quantitatively understand “a larger distortion”? According to Figure 1, does “a larger distortion” indicate that the black dotted ellipse in Figure 1 (a) has a longer major axis than that in Figure 1 (c)? If so, since the five nodes are randomly selected and connected, then what about other red nodes having the same class but not included within the black dotted ellipse?**
>
> A1: We need to clarify that larger distortion does not mean that the major axis is longer. We present a deeper understanding of larger distortion, generalizability regarding Figure 1, and quantitative results.
>
>
> 1. **The understanding of larger distortion.** We believe that distortion indicates that the feature distribution on the new model deviates from the feature distribution of the old model. The two categories can be well separated in the feature distribution of target, but not in the baseline model and lead to incorrect predictions and catastrophic forgetting.
>
> 2. **Generalizability regarding Figure 1.** We have included a new figure to further illustrate the generalizability, it can be seen from Figure R1 of PDF file that the distribution of nodes predicted incorrectly (grey nodes in the figure) also includes some nodes outside the black dotted ellipse. We have supplemented another black dotted ellipse to mark incorrectly predicted nodes to increase clarity.
>
>
> 3. **Quantitative analysis.** We use the structural shift score defined in Equation (4) on Page 4 to quantitatively understand the degree of distortion. The overall structural shift score is 0.1359, the structural shift score for nodes predicted correctly is 0.1431, and the structural shift score for nodes predicted incorrectly is 0.1646.
>
> >**Q2: Do the parameter isolation methods, the replay methods, and the regulation methods mentioned in the Introduction section (from line 28 to line 37), have the same meaning as those mentioned in the Related Work section (from line 83 to line 89)?**
>
> A2: **Although both are classifications of existing methods and use the same classification names, there are still some differences in terms of the perspective of review and the level of detail.** In the Introduction section, we summarize existing methods from the perspective of information preservation as maintaining the previous model or graph data. We provide a detailed introduction to the form of information preservation for each type of method. The Related Work section introduces the definition and relevant papers for each type of method according to the existing classification.
>
>
> >**Q3: From line 216 to line 217, based on Equation 14, since the mean pooling has been applied, what is the motivation and reasoning to introduce the mean squared error (MSE) loss to evaluate global representation gaps?**
>
> A3: Our motivation for using another MSE loss stems from the instantiation of low-frequency global information preservation. MSE loss is employed to measure the differences in global features obtained after pooling. It is more stable in application and has demonstrate the effectiveness of this term in ablation studies.
>
>
> 1. **We have indeed stated that Equation (14) is an instantiation of the low-frequency global information preservation in Equation (11).** As we stated in Lines 184-187 on Page 5, the conclusion drawn from Equation (11) is that to maintain low-frequency global information, it is necessary to reduce the difference in representation of the full replay graph formed by nodes and their multi-order neighbors between new and old models. The two terms of Equation (11) represent the representations of the replay graph on the model, and we instantiate each term through mean-pooling. Additionally, the full graph representation is also meaningful, as it represents the overall graph information contained in each category dimension.
>
> 2. **The reasons for using MSE loss to measure the distribution difference between the representations of old replay graph and new replay graph can be encapsulated in three aspects:**
> * The MSE loss is more stable compared to other distribution difference measure functions [1].
> * The use of MSE loss in Equation (14) does not duplicate that of MSE loss in Equation (13). Equation (13) is an instantiation of the low-frequency local information formed by Equation (10). Both work together to maintain low-frequency graph information.
> * From the ablation study in Table 2, after applying low-frequency local information preservation (Equation 13) on the Reddit dataset, ERGNN-GSIP achieved AP and AF of 84.63\% and -12.09\%. Upon incorporating low-frequency global information preservation (Equation 14), the AP and AF increased significantly to 89.21\% and -6.97\%, representing an improvement of up to 4\%. This confirms the effectiveness of utilizing another MSE loss.
>
> [1] Chaitanya K Joshi et al. On representation knowledge distillation for graph neural networks. TNNLS 2022.

---

> > ### Comment · Reviewer_9jHn · 2024-08-08
> > **Acknowledgment of rebuttal**
> >
> > Thank you for answering my questions regarding the distortion and the motivation of using mean squared error loss. I acknowledge that I have read the rebuttal and have no further questions.

---

> > > ### Author Response · Authors · 2024-08-09
> > > **Thanks for Response**
> > >
> > > We greatly appreciate your comments and recognition. We also look forward to receiving any further suggestions.

---

### Official Review · Reviewer_Za5b · 2024-07-13

**Soundness:** 3
**Presentation:** 3
**Contribution:** 2
**Rating:** 6
**Confidence:** 3

**Summary:**

The paper focuses on the challenge of graph class incremental learning (GCIL), where a model must classify new nodes of emerging classes while retaining knowledge of previously learned classes. The primary issue of GCIL is identified as catastrophic forgetting, where new learning overwrites old knowledge. To address this, the author introduce the concept of information preservation, suggesting that preserving graph information can help mitigate semantic and structural shifts that cause forgetting. The paper proposes the Graph Spatial Information Preservation (GSIP) framework, which preserves both low-frequency (local-global) and high-frequency information in the graph's spatial domain. GSIP aligns old and new node representations, ensuring old graph information is retained. Experimental results show that GSIP significantly reduces forgetting by up to 10% on large datasets compared to existing methods.The framework is shown to be effective across various benchmark datasets.

**Strengths:**

- The proposed method GSIP preserves both low-frequency and high frequency information of a graph, which can alleivate the catastrophic forgetting issue of GCIL.
- The proposed method is proved to be effective across various benchmark datasets.

**Weaknesses:**

- The design of the method, which only capture low and high frequency information is not comprehensive. There are more complicated signals to capture, e.g., many spectral GNNs are designed for that.
- The method is only evaluated on 3 datasets, corafull, arxiv and reddit. It would be great to be evaluated on more datasets.

**Questions:**

- Why the method only consider the low and high frequency information in the graph’s spatial domain? Except for these two frequency, there are still other frequency information worth to be captured, e.g., medium, and more to capture complicated signals.
- Why for the loss function (Equation 19), there is only one scaling factor \alpha for the second loss term?

**Limitations:**

Yes

---

> ### Author Rebuttal · Authors · 2024-08-06
>
> Thanks for your constructive comments!
> >**Q1: The design of the method, which only capture low and high frequency information is not comprehensive. Why does the method only consider the low-/high- frequency information in the graph’s spatial domain?**
>
> A1: Considering the preservation of low-/high- frequency information is essential for addressing the challenge of catastrophic forgetting. We conduct analysis and experiments and conclude that high-frequency information preservation sufficiently calibrates structural shift.
>
>
> 1. **The preservation of low-/high- frequency information has effectively calibrated feature and structural shifts, yielding satisfactory results.**
>
> * Figure 5(c) on Page 8 shows that the semantic and topological shifts are well-calibrated with the final semantic shift score and structural shift score almost reduced to 0.
> * Table 1 on Page 8 shows that catastrophic forgetting is greatly alleviated after using GSIP, the forgetting rate on ERGNN-GSIP is reduced by up to 57.94\%.
>
> 2. **The analysis and experiments of mid-frequency information preservation.**
> * **Method analysis.**
> We conduct a brief analysis of the preservation of mid-frequency information. According to the definition of mid-frequency filtering graph convolutional networks [1][2], the mid-frequency convolution can be expressed as:
> $\mathcal{F}^m={({I} _n-\widetilde{{D}}^{-\frac{1}{2}} \widetilde{{A}} \widetilde{{D}}^{-\frac{1}{2}})}{({I} _n+\widetilde{{D}}^{-\frac{1}{2}} \widetilde{{A}} \widetilde{{D}}^{-\frac{1}{2}})}$. Through a series of analysis, mid-frequency information preserving is defined as:
> $$
> \left \| \Delta{\mathbf{z}^{m} _i}\right\| _2^2 = \left\|\left({z} _i^{old}-\sum _{j \in \mathcal{N}^2 _i} \frac{{z} _j^{old}}{\sqrt{\left|\mathcal{N} _i\right|\left|\mathcal{N} _j\right|}}\right)- \left({{z} _i^{new}-\sum _{j \in \mathcal{N}^2 _i} \frac{{z} _j^{new}}{\sqrt{\left|\mathcal{N} _i\right|\left|\mathcal{N} _j\right|}}
> }\right) \right \| _2^2,
> $$ where $\mathcal{N}^2$ is the second-order neighbors of nodes. The distinction between mid-/high- frequency information preservation lies in that mid-frequency signals calculate the differences between the node and its second-order neighbors.
>
> * **Experimental result.**
> We have added mid-frequency information preservation **(M)** in method, which could yield a slight performance improvement in some cases. However, it does not lead to better performance enhancements or outstanding results. The possible reason is that the preservation of first-order neighbors is sufficient to calibrate structural shift effectively. In the future, we will investigate comprehensive analysis and instantiations for the preservation of complicated signals.
>
> | | |CoraFull| |Arxiv| |Reddit| |
> |:----|:----|:----|:----|:----|:----|:----|:----|
> | | |AP|AF|AP|AF|AP|AF|
> |Unequally|GSIP| **55.32±0.75**| **-2.50±1.13**|63.86±0.85|0.08±0.76|**79.31±0.50**|0.70±0.25|
> | |+M|55.28±0.65|-2.61±1.04| **63.89±0.86**| **0.20±0.68**|79.29±0.67| **0.75±0.52**|
> |Equally (10)|GSIP|63.36±1.13| -7.27±0.82| **61.34±0.77**| **-6.34±0.70**| **64.16±0.37**| **-8.87±0.58**|
> | |+M|**63.48±1.16**|**-7.21±1.06**| 61.28±0.71|-6.50±0.79| 64.08±0.12| -9.41±0.71|
> |Equally (2)|GSIP|**90.74±0.44**| **-3.97±0.40**| **87.41±1.60**|0.13±0.91| **96.25±0.37**| **-0.65±0.64**|
> | |+M|90.65±0.29|-4.01±0.40| 87.34±1.69|**0.18±0.92**|96.11±0.33|-0.82±0.44|
>
>
>
> [1] Jincheng Huang et al. Robust Mid-Pass Filtering Graph Convolutional Networks. WWW 2023.
>
> [2] Haitong Luo et al. Spectral-Based Graph Neural Networks for Complementary Item Recommendation. AAAI 2024.
>
> >**Q2: The method is only evaluated on 3 datasets, Corafull, Arxiv, and Reddit. It would be great to be evaluated on more datasets.**
>
> A2: **We have supplemented results from the newly added Cora and Citeseer datasets.** As shown in Table R1 of PDF file, each dataset has 3 tasks with two categories per task. GSIP achieves significant performance improvements, validating the effectiveness of GSIP. For ERGNN-GSIP, the AP and AF increase by 5.81\% and 9.14\% on Cora, and by 13.64\% and 21.74\% on Citeseer. The AF of SSM-GSIP improves by more than 5\% on two datasets. CaT-GSIP achieves the highest performance in most cases, with the AP approaching the value of Joint.
>
> As for implementation details, we use the same experimental setup as three existing datasets, the experiment includes three hyper-parameters $\alpha_{gip}$, $\beta$, and $\gamma$. On Cora, for ERGNN, SSM, and CaT, the hyper-parameters are [5e-3, 2e1, 2], [1e-1, 1, 5e-2], and [1e-2, 1e3, 1e2] respectively. On Citeseer, for ERGNN, SSM, and CaT, the hyper-parameters are [5e-2, 5e3, 1e1], [1e-3, 1e2, 5], and [1e-3, 1, 1e-3] respectively. The number of budgets $\|\mathcal{M}\|$ is 10\% of the number of budgets for the existing datasets.
>
>
> >**Q3: Why for the loss function (Equation 19), there is only one scaling factor $\alpha$ for the second loss term?**
>
> A3: **We have added a new hyper-parameter $\alpha_{gip}$ to adjust the weight of $\mathcal{L} _{gip}$** and Equation (19) is modified to $\mathcal{L}=\mathcal{L} _{nc}+\alpha _{replay}\mathcal{L} _{replay}+\alpha _{gip}\mathcal{L} _{gip}$.
>
> **We analyze the impact of $\alpha_{gip}$ on AP and AF of ERGNN, SSM, and CaT across three datasets with increments of 2 in Figure R2 of PDF file.** For ERGNN/SSM/CaT, $\alpha_{gip,1}$ are set to [1, 1, 0.1], [0.01, 0.01, 0.01], and [0.1, 0.01, 0.01] for three datasets. It can be observed that the performance change is not as significant with the variation of $\alpha_{gip}$ on SSM-GSIP and CaT-GSIP. However, different $\alpha_{gip}$ have a greater impact on performance with ERGNN-GSIP. The possible reason is that ERGNN selects representative nodes for replay, which may cause class imbalance and discard structure. For ERGNN, SSM, and CaT, the optimal hyper-parameters $\alpha_{gip}$ on three datasets are [50, 10, 1], [0.1, 0.1, 0.1], and [1, 0.5, 0.5].

---

> > ### Comment · Reviewer_Za5b · 2024-08-10
> > **increase rating from 5 to 6**
> >
> > Thanks for answering my concerns. I have increased the rating of the paper from 5 to 6 accordingly.

---

> > > ### Author Response · Authors · 2024-08-11
> > > **Thanks for Response**
> > >
> > > Thank you for increasing your rating and your support for the paper. Please let us know if you have any additional questions or concerns.

---

### Author Rebuttal · Authors · 2024-08-06

We extend our gratitude for the valuable feedback and insightful suggestions provided by all the reviewers. We have diligently addressed the questions and suggestions raised during the official review process and have provided comprehensive response to the reviewers in the corresponding rebuttals.

We are delighted to be recognized for our efforts in this research. We would like to thank Reviewer Za5b for acknowledging the motivation and effectiveness of our model in avoiding catastrophic forgetting on graph class incremental learning. We also extend our appreciation to Reviewer 9jHn for their strong recognition of our novelty, theoretical contributions, and comprehensive experiments. Furthermore, we appreciate Reviewer BwJ9's acknowledgment of our theoretical foundations, empirical improvements, and quantitative analysis. Additionally, we are grateful to Reviewer T3gk for their strong endorsement of the motivation, theoretical contributions, and effectiveness of our work.

We have conducted analysis and experiments on mid-frequency information preservation, validating our results on new datasets and analyzing the weights of graph information preservation loss.
We have advanced an understanding of larger distortion and performed a quantitative analysis. We have clarified the differences between the overview of recent papers in the introduction and related work sections and elucidated the motivation and reasons for using MSE loss to measure the discrepancy in global representations.
We have explained certain terms before their use and modified some notations. The rationale behind the definition of the structural shift score has been expounded. We have elucidated the reasons for modeling from low-/high- frequency information preservation.
We present the reasons for not employing high-frequency global information from the perspective of method design and experimental outcomes, explain the existing task division, list detailed experimental details and hyperparameters, and conclude with an analysis of the replay loss weights.

**We also provide a rebuttal PDF attached below in this global response section to exhibit additional results.** We have supplemented Table R1 with results from the newly added Cora and Citeseer datasets. Figure R1 illustrates the visualization of node representation on node feature shift and its generalizability. Table R2 shows the hyperparameter settings used in the experiments. In Figure R2 and Figure R3, we analyze the effect of loss weights of graph information preservation loss and replay loss on performance.

---

### Decision · Program_Chairs · 2024-09-25

**Decision:**

Accept (poster)

**Comment:**

This paper proposes a graph spatial information preservation (GSIP) framework to alleviate catastrophic forgetting in graph class incremental learning (GCIL), with a focus on theoretically investigating what matters in preserving the information from the old classes. The authors have done a good job in the theoretical analysis in the topic of graph class incremental learning, which appears to be a nice addition to the community. The proposed method has also been shown to be effective across various benchmark datasets. Overall, the reviews are positive, and I recommend accepting the paper. However, there are some comments on the clarity of the presentation, and I encourage the authors to carefully revise the final version.